# Aerosol type classification with machine learning techniques applied to multiwavelength lidar data from EARLINET

Ana del Águila[1,2], Pablo Ortiz-Amezcua[1,2], Siham Tabik[3], Juan Antonio Bravo-Aranda[1,2], Sol Fernández-Carvelo[1,2] and Lucas Alados-Arboledas[1,2]

[1]Andalusian Institute for Earth System Research (IISTA-CEAMA), 18006 Granada, Spain
[2]Department of Applied Physics, University of Granada, 18071 Granada, Spain
[3]Department of Artificial Intelligence, University of Granada, 18071 Granada, Spain

*Correspondence to*: Ana del Águila (anadelaguila@ugr.es)

**Abstract.** Aerosol typing is essential for understanding atmospheric composition and its impact on the climate. Lidar-based aerosol typing has been often addressed with manual classification using optical property ranges. However, few works addressed it using automated classification with machine learning (ML) mainly due to the lack of annotated datasets. In this study, a high-vertical-resolution dataset is generated and annotated for the University of Granada (UGR) station in Southeastern Spain, which belongs to the European Aerosol Research Lidar Network (EARLINET), identifying five major aerosol types: Continental Polluted, Dust, Mixed, Smoke and Unknown. Six ML models - Decision Tree, Random Forest, Gradient Boosting, XGBoost, LightGBM and Neural Network- were applied to classify aerosol types using multiwavelength lidar data from EARLINET, for two system configurations: with and without depolarization data. LightGBM achieved the best performance, with precision, recall, and F1-Score above 90% (with depolarization) and close to 87% (without depolarization). The performance for each aerosol type was evaluated and dust classification improved by ~30% with depolarization, highlighting its critical role in distinguishing aerosol types. Validation against independent datasets, including a smoke case and a Saharan dust event, confirmed robust classification under real and extreme conditions. Compared to NATALI, a neural network-based EARLINET algorithm, the approach presented in this work shows improved aerosol classification accuracy, which emphasize the benefits of using high-resolution multiwavelength lidar data from real measurements. This highlights the potential of ML-based methods for robust and accurate aerosol typing, establishing a benchmark for future studies using multiwavelength lidar at high-resolution data from EARLINET.

## 1 Introduction

The accurate and automated classification of aerosol types is crucial for understanding atmospheric composition and their interactions with the climate system. Aerosols originate from several sources and influence the Earth's radiative balance directly, by absorbing or scattering radiation, and indirectly, through their role in cloud formation and precipitation (IPCC, 2023). Moreover, different aerosol types have distinct effects in such radiative balance (Matus et al., 2019). Thus, accurate aerosol classification is highly relevant for improving climate models and enhancing the accuracy of satellite data retrievals (Chen et al., 2024).

Multiwavelength lidars provide atmospheric vertically resolved information on aerosol optical properties (namely backscattering and extinction coefficients), which reveal important details about particle size, shape and composition (Ortiz-Amezcua et al., 2017; Benavent-Oltra et al., 2019; 2021; Soupiona et al., 2019; 2020). Thus, this information enables the inference of aerosol types, providing a deeper understanding of their role in the atmosphere.

Aerosol typing schemes for lidar systems are generally based on observational results which attribute a certain type of aerosol to a specific range of optical properties. The most common optical properties used in the literature for aerosol typing are intensive properties such as the lidar ratio and the particle linear depolarization ratio at either 355 or 532 nm (Groß, et al., 2013; 2014; Navas-Guzman et al., 2013; Illingworth et al., 2015; Soupiona et al., 2020). Despite depolarization products require calibration and uncertainty assessment (Bravo-Aranda et al, 2016; Freudenthaler, 2016; Belegante et al. 2018), their critical role in the aerosol typing process justifies the effort. Moreover, the intensive properties are type-dependent and thus provide a higher level of information for classifying aerosols. Other authors include additional intensive properties to their classification schemes such as the Ångström exponent (Baars et al., 2017) or the color ratio for different pairs of wavelengths (Groß, et al., 2013), as well as extensive properties such as the backscatter coefficient (Baars et al., 2017; Kim et al., 2018).

One of the major challenges in aerosol typing from lidar measurements is the variability in optical property ranges across studies, which mostly depend on the location. While the ranges are generally similar for the same aerosol type, the associated error can vary significantly among studies (Nicolae et al., 2018). To address this, datasets such as DeLiAn (Floutsi et al., 2023) compile lidar-derived intensive optical properties from ground-based observations, which provide typical values for different aerosol types. However, the global datasets might have limitations in capturing detailed vertically resolved aerosol properties, which are important for achieving an accurate aerosol classification. Given the variability and complexity of aerosol properties, machine learning (ML) offers a robust approach to overcome these challenges and to improve the accuracy and consistency in the aerosol typing task.

In the recent years, there is a focus on finding automatic aerosol classification schemes by either applying statistical methods to lidar-derived intensive properties (Floutsi et al., 2024), by the source of aerosols based on the geographical region information (Mylonaki et al., 2021), by applying supervised learning techniques such as clustering analysis (Papagiannopoulus et al., 2018) or by applying artificial neural network-based techniques (Nicolae et al., 2018). To the best of our knowledge, no previous studies have evaluated different ML algorithms for aerosol typing using lidar optical properties. To fill the gap, this study provides an assessment of various ML algorithms applied to both extensive and intensive lidar properties for aerosol classification. Using data from the the European Aerosol Research Lidar Network (EARLINET, Pappalardo et al., 2014), specifically from the University of Granada (UGR) station in Spain, the aerosol layers are automatically detected and the extensive and intensive properties are computed at high vertical resolution. The use of Aerosol, Clouds and Trace Gases (ACTRIS)/EARLINET validated data ensures robustness and consistency in the testing of the proposed ML methods. By evaluating several ML models, we have identified the most accurate ML algorithm for aerosol typing, improving on current state-of-the-art methods and thus establishing a benchmark for future research on this field.

This work is organized into the following sections: Section 2 provides an overview of the methodology employed, from the preprocessing of the lidar data, going through the reference dataset design and finally to annotating the data. In addition, the ML methods applied are explained, including details of the feature selection and hyperparameter tuning process. Section 3 presents the results, comparing the performance of different ML models and configurations: with and without depolarization. Also, a validation of the ML model and a comparison with NATALI model is presented. Section 4 discusses the findings,
novelties and potential applications of the proposed approach. In Section 5 the conclusions of the work are presented, offering an overview of the future research directions.

## 2 Methods

### 2.1 Reference dataset: data acquisition, processing and annotation

The multiwavelength lidar data used in this work were collected from the ACTRIS-EARLINET database (https://earlinet.org)
for the UGR station, which is located in Granada (Spain) at the Andalusian Institute for Earth System Research (37.16ºN, 3.6ºW, 680 m a.s.l.) and is part of the Andalusian Global Observatory of the Atmosphere (AGORA). The UGR station also belongs to the ACTRIS research infrastructure (Laj et al., 2024). The city of Granada is located in the southeastern part of the Iberian Peninsula, where its local aerosol loading and meteorological characteristics are strongly influenced by its urban nature and by the complex-terrain of Sierra Nevada Mountain area (del Águila et al., 2018; 2024). The major external source of
aerosols in this region is North Africa, which leads to frequent Saharan dust events (Guerrero-Rascado et al., 2008; 2009; Cazorla et al., 2017; Soupiona et al., 2020). In addition, biomass burning aerosols transported from the Iberian Peninsula, North Africa and North America are frequent (Alados-Arboledas et al., 2011; Ortiz-Amezcua et al., 2017; Titos et al., 2017). The conceptual overview of the methodology of this work is shown in Fig. 1 and described in detail in the following sections.

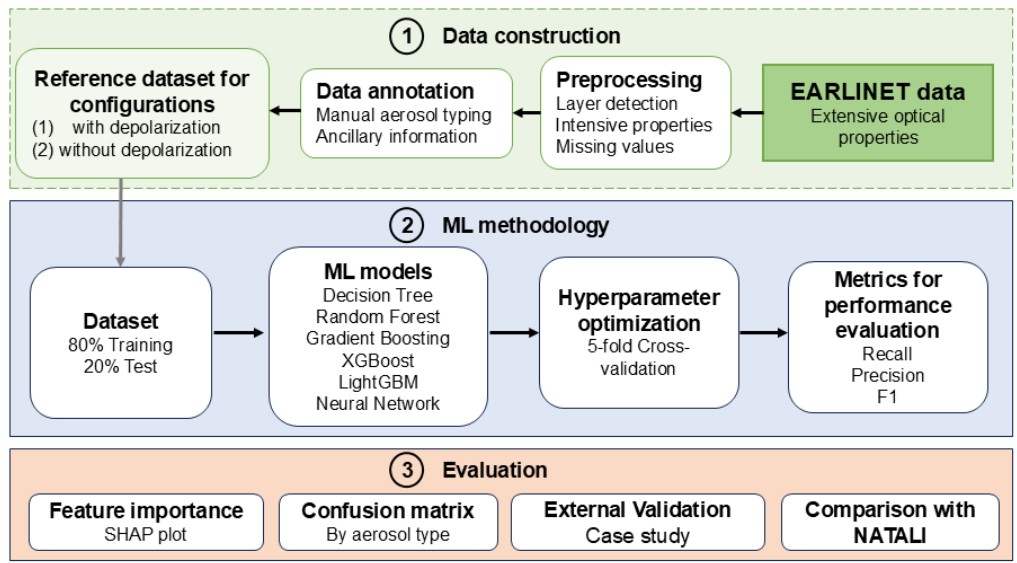

**Figure 1.** Conceptual diagram of the methodology developed for aerosol typing.

EARLINET database includes information on the vertical profiles of particle backscatter ($\beta_{par}$), extinction coefficient ($\alpha_{part}$) and linear particle depolarization ratio ($\delta_{part}$) at several wavelengths with high vertical resolution (several meters) and time resolution (typically, between 30 min to a few hours), measured by multiwavelength elastic or Raman lidar systems located in different stations across Europe. The UGR station was equipped with the MULHACEN Raman lidar (LR331D400, Raymetrics S.A.) from 2005 until 2020 (e.g. Granados-Muñoz et al., 2012), and provided profiles of $\beta_{par}$ at 355, 532 and 1064 nm, $\alpha_{part}$ at 355 and 532 nm, and $\delta_{part}$ at 532 nm with a vertical resolution of 7.5 m for half-hour profiles.

A well-characterized dataset of MULHACEN particle optical profiles and their respective errors for a period between 2012 and 2015 was selected and obtained from EARLINET database for this analysis. Then, we computed the following intensive properties: the extinction Ångström exponent (AE) for the pair 355-532 nm (Eq.1); the backscatter Ångström exponent (color index, CI) for the pairs 355-532 nm, 532-1064 nm (Eq. 2); the backscatter ratio (color ratio, CR) for the pairs 355-532 nm and 532-1064 nm (Eq. 3) and the lidar ratio (LR) at 355 and 532 nm (Eq. 4). In addition, the respective uncertainties of the derived optical products were also computed.

$$AE = -\frac{\ln(\alpha_{\lambda_1}/\alpha_{\lambda_2})}{\ln(\lambda_1/\lambda_2)} \tag{1}$$

$$CI = -\frac{\ln(\beta_{\lambda_1}/\beta_{\lambda_2})}{\ln(\lambda_1/\lambda_2)} \tag{2}$$

$$CR = \frac{\beta_{\lambda_1}}{\beta_{\lambda_2}} \tag{3}$$

$$LR = \frac{\alpha_{\lambda_1}}{\beta_{\lambda_1}} \tag{4}$$

The aerosol layer boundaries (top and bottom) were determined following the methodology explained in Nicolae et al. (2018), which makes use of the aerosol backscatter coefficient at 1064 nm and applies the gradient method (Belegante et al., 2014). Thus, the first and second derivatives of $\beta_{1064}$ are computed with a third order Savitzky-Golay filter, in order to obtain the inflexion points of the second derivative that delimitate the boundaries of the aerosol layer for each profile. The window size used was set to 700 m with a minimum height of 300 m and a signal-to-noise ratio of 5.

We calculated the average intensive parameters for each aerosol layer. These average values were then assigned uniformly across the entire layer. As a result, the database includes two representations of the intensive properties: one at the lidar resolution, where the properties are height-resolved, and another where the average intensive properties are assigned to each aerosol layer. In the latter case, the same average value is repeated across the height of the layer, i.e., maintaining the lidar resolution (height-resolved) for consistency. The use of height-resolved data provides a detailed representation of the vertical distribution of aerosol properties, while the layer-averaged values, repeated across all height levels within each layer, provide a consistent profile representative of the overall aerosol type. This combination allows ML models to learn from both fine-scale variability and averaged layer characteristics, which is particularly beneficial for improving prediction accuracy and capturing more complex aerosol variations within layers.

For annotating the aerosol types of the reference database, we applied a manual labelling to 416 aerosol layers. Based on the literature (Groß et al., 2013; 2014; Navas-Guzman et al., 2013; Illingworth et al., 2015; Soupiona et al., 2020), we assigned a single type of aerosol to each aerosol layer at average resolution according to certain ranges of the calculated average intensive properties, as described in Table 1. These properties range criteria were a first attempt for aerosol classification. This initial approach was followed by a thorough review of each vertical profile and its optical properties by experts. Corrections were made to the aerosol type in cases of misclassification identified during the first attempt. Finally, to ensure accurate classification, we employed ancillary information to verify the aerosol type assigned to each layer. Therefore, further analysis was carried out by running HYSPLIT backtrajectories (Stein et al., 2015) and NAAPS model (Lynch et al., 2016) to support the labelling. For the cases where the aerosol type was not clear enough, we computed the backtrajectories with the HYSPLIT model, taking as the starting point of the altitude for the backtrajectory analysis the one of each identified aerosol layer. Thus, for uncertain aerosol layer types, we ran the model for 5 days in advance to trace the air mass origin for the altitude corresponding to each layer. In addition, we assessed temporal consistency by verifying that the aerosol types between layers remained consistent from subsequent profiles.

**Table 1**. Indicative ranges of lidar properties for manual aerosol typing at UGR station, adapted for this study and based on typical ranges in the literature, were used as an initial reference for the manual labelling process. This first attempt for aerosol classification was followed by a thorough review of each vertical profile and corrections were made when necessary to ensure accurate classification. Lidar ratios (LR) are expressed in steradians (sr).

| Aerosol type | Properties range criteria |
| --- | --- |
| Clean Cont. | $22 \leq LR_{355} \leq 36$ & $0.02 \leq \delta_{part} \leq 0.06$ |
| Volcanic | $30 \leq LR_{532} \leq 60$ & $0.33 \leq \delta_{part} \leq 0.46$ |
| Smoke | $26 \leq LR_{532} \leq 100$ & $0.001 \leq \delta_{part} \leq 0.14$ |
| Dust | $32 \leq LR_{532} \leq 71$ & $0.1 \leq \delta_{part} \leq 0.32$ |
| Mixed | $32 \leq LR_{532} \leq 71$ & $0.1 \leq \delta_{part} \leq 0.2$ |
| Cont. Polluted | $42 \leq LR_{532} \leq 81$ & $0.025 \leq \delta_{part} \leq 0.07$ & $1.7 \leq CR_{355,1064} \leq 2.7$ & $0.4 \leq AE_{355,532} \leq 1.6$ |
| Unknown | Else |

Finally, the aerosol types found at UGR station are: Continental polluted, Dust, Smoke, Mixed and Unknown. The inclusion of the 'Unknown' class is a novel approach in this field, as it represents cases where the aerosol type cannot be definitively classified. In contrast, methods like NATALI assign an equivalent 'unknown' label when the neural network fails to identify a final class, either due to error thresholds being exceeded or the inability to compute intensive properties. The 'Mixed' aerosol type corresponds to two or more aerosol components (Li et al., 2016) such as dust and smoke, or dust and continental polluted.

Once the aerosol types were certain for each layer, we applied the same label for each altitude at the lidar height-resolution. Finally, the height-resolved labels from the reference dataset are used for comparison with the outputs of each ML method. The overview of the approach to design the reference dataset is shown in Fig. 2.

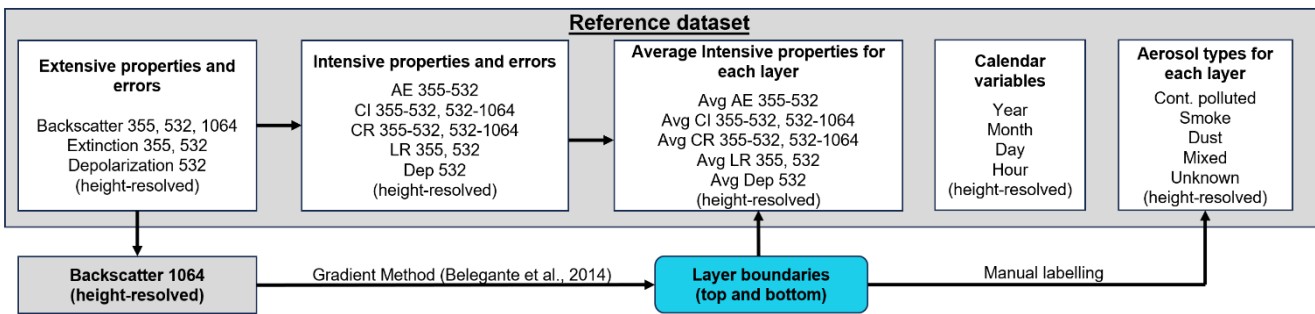

**Figure 2.** Diagram of the reference dataset design.

## 2.2 Missing values

Missing values can be addressed using different methods, ranging from simple techniques like mean imputation to more advanced approaches such as Long Short-Term Memory (LSTM)-based modelling (Alabadla et al., 2022). In this study, the missing values of each variable at vertical resolution have been imputed with the median value of each profile. Let us name each profile by $j = 0,1,2, \dots, n-1$ and define the height discretization $z_i^j = z_0^j + i\Delta z, \ i = 0,1,2, \dots, m-1$, where $n, m$ are

positive integer values. $\Delta z$ represents the step in altitude, i.e., the vertical spatial resolution, and $z_i^j$ denotes that $z_0^j$ depends on the profile. Thus, each variable $p$ of the reference dataset complies with: $p_i^j \coloneqq p(z_i^j, j)$ and the sequence of values for all the profiles be written as $\left(p_i^j\right)_{i,j\geq 0}$. Finally, the imputation of the missing values has been performed as follows:

$$\left(p_i^j\right)_{i\geq 0} = \begin{cases} p_i^j \text{ if it exists} \\ \text{median}\left(p_i^j\right)_{i\geq 0} \end{cases}$$

Which means that for a fixed profile we assign for each height $z_i^j$ the value $p_i^j$ if it exists and the median of the existing values

for that profile if it does not exist.

### 2.3 Machine learning models for classification

We have evaluated different supervised machine learning (ML) models for two configurations of the reference database: (1) with depolarization and (2) without depolarization. For each configuration, the reference dataset was divided into training (80%) and testing (20%). We assessed the performance of Decision Trees (Quinlan, 1986), Random Forest (Breiman, 2001),

LightGBM (Ke et al., 2017), Gradient Boosting, XGboost (Chen and Guestrin, 2016) and Neural Networks (Li et al., 2022). Below there is a summary of each ML model applied in this study:

- Decision Tree: this model was used to build a simple baseline classifier by splitting the dataset into several feature thresholds to predict the aerosol type. The key hyperparameters of these models are the maximum depth (max_depth) of the trees and the minimum samples of the split (min_samples_split).

- Random forest: the method builds multiple decision trees and merge them together to get a more accurate and stable prediction and controls overfitting by averaging multiple deep decision trees, trained on different parts of the same training set. To find the best configuration for this method, we have varied the number of trees (n_estimators) and the maximum depth (max_depth) of the trees.

- Gradient boosting: the method builds an additive model incrementally to allow optimizing arbitrary differentiable loss functions. The key hyperparameters of this method include the number of boosting stages (n_estimators) and the learning rate (learning_rate).

- XGBoost: the method consists of an implementation of gradient boosting decision trees for speed and performance. The method is tuned by the number of boosting rounds (n_estimators), the learning rate (learning_rate) and the maximum tree depth (max_depth).

- LightGBM: the method is a high-performance gradient boosting framework which uses tree-based learning algorithms. The method is adjusted by the number of leaves in a tree (num_leaves), the learning rate (learning_rate), and the number of trees (n_estimators).

- Neural network (MLPClassifier): It captures complex relationships in data, the neural network (NN) has been configured with different architectures (hidden_layer_sizes), regularization terms (alpha), and initial learning rates (learning_rate_init).

The ML models (ML_models) were chosen to solve the classification problem framed as the aerosol typing for each vertical profile and layer as follows:

$$\hat{c}_i^j = \text{ML\_model}[p_{i,k}^j]$$

Where $p_{i,k}^j$ represents the reference dataset, with all the variables $k = 0, 1, ..., l-1$ from the dataset (Fig. 2), and $\hat{c}_i^j$ is the predicted class for each layer at height resolution. For all ML models, each point of the aerosol layer was considered as a discrete observation for supervised learning. Therefore, the methodology automatically classifies the aerosol type for vertically resolved data, given the detected aerosol layers. The ML models have been implemented using libraries like scikit-learn (Pedregosa et al., 2011) in Python. This approach enables the automatic classification of aerosol types in a vertical column, providing high vertical resolution information for atmospheric studies.

## 2.4 Feature importance analysis

In order to understand which features (variables) of the ML models have more influence to predict the aerosol type, we have performed feature importance analysis by calculating the importance score for each feature with the Shapley additive explanations (SHAP) method (Lundberg & Lee, 2017). This approach makes use of game theory to quantify the features with

respect to the output model. SHAP assigns each feature an importance value for a particular prediction (Lundberg and Lee, 2017), helping the correct interpretation of the output of a prediction model. Thus, the SHAP method provides an interpretation scheme for the ML models and, specifically, the use of SHAP to lidar data helps evaluating the significance of the lidar properties into achieving the desired aerosol typing.

## 2.5 Hyperparameters optimization

ML methods are parametrized with the named hyperparameters. To ensure that the ML models were neither overfitted nor underfitted, we performed hyperparameter tuning using the GridSearchCV function from Python's scikit-learn library (Pedregosa et al., 2011), with five-fold cross-validation (Breiman and Spector, 1989). The tuning process was applied to all models, including Decision Tree, Random Forest, Gradient Boosting, XGBoost, LightGBM, and Neural Networks, with a focus on optimizing recall for the multi-class classification problem. The selected ranges of hyperparameters were based on similar studies of the literature (e.g. Philippus et al., 2024; see Table 3 for evaluated values). For the decision tree, we varied the maximum tree depth (max_depth) and the minimum number of samples required to split a node (min_samples_split). In the random forest, we adjusted both the number of trees (n_estimators) and the maximum depth (max_depth). Similarly, for gradient boosting and XGBoost, hyperparameters such as the number of boosting rounds (n_estimators), learning rate (learning_rate), and tree depth (max_depth) were optimized. LightGBM was fine-tuned by varying the number of leaves (num_leaves), learning rate, and the number of boosting stages. The Neural Network model was configured with different architectures by adjusting the number of hidden layers and neurons (hidden_layer_sizes), learning rate (learning_rate_init), and regularization parameter (alpha). All models were evaluated using GroupKFold to ensure consistent cross-validation across different groups (layers) of the training dataset. GroupKFold is a cross-validation strategy in which the data is split into training and testing sets while ensuring that the same group is not represented in both sets. The detailed explanation is included in the scikit-learn documentation on GroupKFold (https://scikit-learn.org/stable/modules/generated/sklearn.model_selection.GroupKFold.html). The best hyperparameters for each model were identified based on the weighted accuracy, and the optimal configurations were subsequently used to evaluate the models on the testing dataset. Thus, an optimal set of hyperparameters for each method, will result in the best configuration which minimizes the corresponding loss function.

## 2.6 Performance evaluation

Once the ML models were applied, we evaluated their performance using the reference dataset, which was divided into training (80%) and testing (20%) sets. The 80/20 split is a commonly adopted practice in ML, as it provides a good balance between the amount of data available for training and a sufficiently large test set for reliable performance evaluation (Kohavi, 1995). This proportion is usually considered to offer an optimal balance, ensuring that the model is evaluated on a sufficiently diverse test set and maintains the integrity of the evaluation process (Zhou, 2017). The division was done using stratified sampling to maintain the proportional distribution of aerosol classes across both sets. To ensure that data from the same aerosol layers did

not leak into both the training and testing phases, we applied group-based segregation using GroupShuffleSplit, which ensured that data points from the same aerosol layer were kept together either in the training or testing sets. GroupShuffleSplit is a cross-validation strategy in which the data is split into training and testing sets while ensuring that the same group is not present in both sets, similar to GroupKFold. However, unlike GroupKFold, GroupShuffleSplit randomly shuffles the groups and 
allows for more flexible splits, where each group is assigned to either the training or testing set ([https://scikit-learn.org/stable/modules/generated/sklearn.model_selection.GroupShuffleSplit.html](https://scikit-learn.org/stable/modules/generated/sklearn.model_selection.GroupShuffleSplit.html) ). This approach was essential to avoid data leakage, as our dataset includes repeated measurements from the same aerosol layers.

The metrics to evaluate the performance of the classification of the ML models are based on the proportion of correctly predicted classes (both true positives and true negatives) of the total classes. To account for the imbalance in the distribution 
of aerosol classes, weighted metrics were used, which assign weights to each class, that are proportional to their representation in the reference dataset. Thus, the following metrics are evaluated:

$$\text{Precision} = \frac{\text{TP}}{\text{TP} + \text{FP}}$$

$$\text{Recall} = \frac{\text{TP}}{\text{TP} + \text{FN}}$$

$$\text{F1} = 2 \cdot \frac{\text{Precision} \cdot \text{Recall}}{\text{Precision} + \text{Recall}}$$

where TP is True Positives, TN is True Negatives, FP is False Positives and FN is False Negatives. Therefore, precision indicates the model's ability to avoid false positives; recall shows the model's ability to identify true positives and the F1-score captures the balance between accuracy and recall, which is especially useful when the distribution of classes is uneven. The weights used in the calculation of the weighted metrics (such as weighted precision, recall, and F1-score) are based on the relative support of each class in the reference dataset. That is, each class is weighted proportionally to its number of instances
in the dataset and is calculated as:

$$M_{weighted} = \sum_{i=1}^{C} \frac{n_i}{N} \cdot M_i$$

Where $M$ is the metric (precision, recall or F1-score), $C$ is the number of classes (aerosol types), $n_i$ is the number of samples in class $i$, $N$ is the total number of samples, and $M_i$ is the metric for each class $i$. These weighted metrics were used specifically to mitigate the influence of class imbalance in the evaluation. To ensure robustness and reliability, these metrics were 
calculated for the final testing dataset. The models' ability to generalize was further assessed by analyzing how they performed under different aerosol layer configurations, with special attention paid to handling class imbalances in the dataset.

# 3 Results

## 3.1 Reference dataset overview

In this work, we use high-resolution vertical data for the aerosol typing task, rather than using averaged values for each aerosol layer. The vertical resolution data included in EARLINET is directly used, ensuring a robust and detailed representation of aerosol properties which have passed quality assurance filters. This inclusion of all data is one of the major assets of this study, because it allows the application of the resulting ML-model automatically even for datasets including noise or high associated errors (e.g. due to low aerosol concentrations). Usually, the aerosol classification methods ignore those data, but in our approach the associated aerosol layer are classified as Unknown. Therefore, this class represents layers where the aerosol type could not be identified due to significant errors or non-physical meaning. The Unknown class is included as an additional category into the ML models. Hence, they are trained to recognize and correctly predict these challenging cases, as will be shown in the following sections.

Figure 3 provides an overview of the reference dataset, showing the distributions of the main variables for two categories: (1) the Unknown class and (2) the rest of the aerosol types, including Dust, Mixed, Smoke and Continental Polluted. The Unknown class is characterized by backscatter values clustered near 0 and extinction coefficients concentrated between 0 and 50 Mm$^{-1}$. These low values impact on the derived intensive properties such as the LR, which shows higher values (>150 sr), as well as the CR and CI. Regarding the classified aerosol types (Dust, Mixed, Smoke, and Continental Polluted), the distributions of intensive properties fall within the expected ranges. Backscatter coefficients range from 0 to 3 Mm$^{-1}$sr$^{-1}$ at 355 nm and from 0 to 2.5 Mm$^{-1}$sr$^{-1}$ at 532 nm and 1064 nm. Extinction coefficients span from 0 to 200 Mm$^{-1}$ for both wavelengths, while the Angstrom Exponent (AE) ranges from -2 to 2.5. The CI values vary from -0.5 to 2.5, and LR ranges from 0 to 200 sr for both wavelengths. Depolarization values range from 0 to 32%, with approximately 80% of values below 20%, reflecting predominantly spherical particles. Notably, the ranges of these intensive properties align with those reported in previous studies (e.g., Nicolae et al., 2018). However, in some cases, the ranges are slightly broader due to the nature of high-resolution data compared to the averaged data. In contrast, the high-resolution approach provides more detailed information of aerosol properties, improving the ability of ML models to effectively classify the aerosol types.

Figure 4 shows the histograms of the altitude for each aerosol type. Each color represents a different aerosol type, illustrating the variation in altitude aerosol layers across the different types. The Unknown class is the most numerous and prevalent type over a wide distribution across all altitudes. The Smoke type is mostly comprised between 1300 and 6000 m. The Dust type shows a significant presence with predominancy at altitudes from 2300 to 5500 m. The Mixed type shows a similar distribution to dusty aerosols but reaches up to 4300 m. Finally, the Continental Polluted aerosol type show a narrow distribution over 1500 to 3000 m.

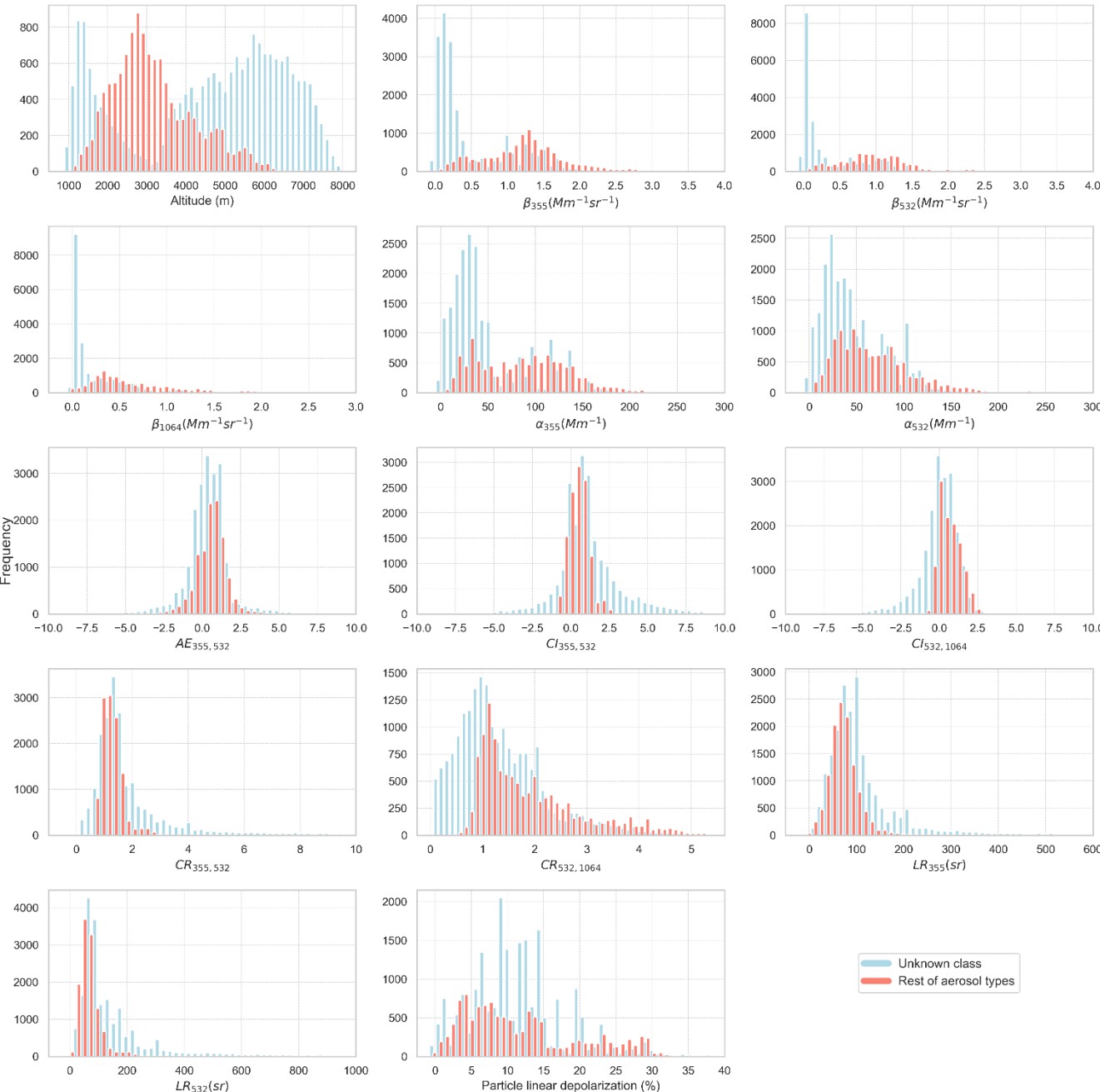

**Figure 3.** Distributions of the key extensive and intensive properties derived from the multiwavelength lidar for the Unknown class (blue) and the rest of the aerosol types (red). The distributions of the properties include the backscatter coefficients ($\beta_{355}, \beta_{532}, \beta_{1064}$), color indexes ($CI_{355,532}, CI_{532,1064}$), color ratios ($CR_{355,532}, CR_{532,1064}$), lidar ratios ($LR_{355}, LR_{532}$) and particle linear depolarization ratio of the reference dataset.

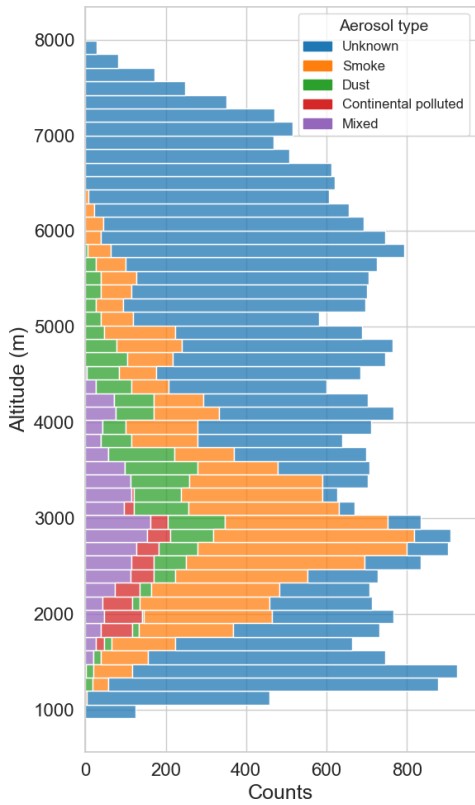

**Figure 4.** Distribution of altitudes for the different aerosol types of the reference dataset.

Table 2 lists the number of manually classified layers and the number of counts for each class. Thus, 416 individual aerosol
layers are identified, with the predominant aerosol types being, in order of frequency: Smoke, Dust, Mixed and Continental
Polluted, which sum 145 classified aerosol layers for the four aerosol types and 11579 counts or heights. In contrast, 271 layers
were classified as Unknown, reflecting the inherent challenges of manual labelling when aerosol types are difficult to discern.
These Unknown layers often correspond to non-physical values or exhibit high associated errors, yet they remain an integral
part of the dataset, thereby underlining the complexity of accurately classifying certain aerosol layers.

**Table 2.** Number of manually classified aerosol layers by aerosol type and their corresponding counts for the reference dataset.

| Aerosol type | # layers | # of counts |
|---|---|---|
| Smoke | 91 | 7044 |
| Dust | 30 | 2257 |
| Mixed | 19 | 1649 |
| Continental Polluted | 5 | 629 |
| Unknown | 271 | 20027 |
| Total | 416 | 31606 |

### 3.2. Model design and performance

### 3.2.1 Hyperparameter optimization

The strategy for obtaining the optimal hyperparameters for each ML model was to use the module GridSearchCV as explained in Section 2.5, which performs an exhaustive search over a specified parameter grid. Thus, the approach ensures that the chosen configuration is the best configuration in terms of performance and minimizes overfitting. Table 3 summarizes the set of hyperparameters evaluated and provides the optimal hyperparameters for each ML model and for both configurations: (1) with depolarization and (2) without depolarization. Therefore, depending on each ML model, the hyperparameters that influence the performance are different. Furthermore, we observe that the optimal parameters for the different ML models also change with the configurations.

**Table 3.** Ranges of hyperparameters evaluated for each ML model and optimal hyperparameters selected for the different configurations.

| ML model | Hyperparameters evaluated | Configuration: with depolarization | Configuration: without depolarization |
|---|---|---|---|
| Decision Tree | max_depth: None, 10, 20, 30 <br> min_samples_split: 2, 10, 20 | max_depth: None <br> min_samples_split: 2 | max_depth: None <br> min_samples_split: 2 |
| Random Forest | max_depth: None, 10, 20 <br> n_estimators: 100, 200 | max_depth: 10 <br> n_estimators: 100 | max_depth: 10 <br> n_estimators: 200 |
| Gradient Boosting | learning_rate: 0.01, 0.1, 0.2 <br> n_estimators: 100, 200 | learning_rate: 0.1 <br> n_estimators: 200 | learning_rate: 0.2 <br> n_estimators: 100 |
| XGBoost | learning_rate: 0.01, 0.1, 0.2 <br> max_depth: 3, 6, 9 <br> n_estimators: 100, 200 | learning_rate: 0.2 <br> max_depth: 9 <br> n_estimators: 200 | learning_rate: 0.2 <br> max_depth: 6 <br> n_estimators: 200 |
| LightGBM | learning_rate: 0.01, 0.1, 0.2 <br> n_estimators: 100, 200 <br> num_leaves: 31, 50, 100 | learning_rate: 0.2 <br> n_estimators: 100 <br> num_leaves: 31 | learning_rate: 0.2 <br> n_estimators: 100 <br> num_leaves: 31 |
| Neural Network | alpha: 0.0001, 0.001, 0.01 <br> hidden_layer_sizes: (64, 64), (128,64), (128, 128) <br> learning_rate_init: 0.001, 0.01 | alpha: 0.001 <br> hidden_layer_sizes: (64, 64) <br><br> learning_rate_init: 0.01 | alpha: 0.01 <br> hidden_layer_sizes: (128, 128) <br><br> learning_rate_init: 0.01 |

### 3.2.2 Evaluation of different ML models

Six ML models were evaluated on the 416 layers from the reference dataset for the period 2012-2015, assessing all metrics on the test dataset for two configurations: (1) with depolarization and (2) without depolarization. The best hyperparameters for each ML model were selected in order to evaluate the different ML models (Table 3). Figure 5 shows the weighted metrics of recall, precision and F1-Score. In general, the ML models that incorporate depolarization data demonstrate significantly higher performance compared to those without depolarization, except for the neural network. These results are in line with those reported by Nicolae et al. (2018), where a neural network (NN)-based classification algorithm was considered for cases with and without depolarization. In that study, the inclusion of depolarization led to a higher number of aerosol types. However, this study shows that for the NN model, the configuration without depolarization depicts higher performance. This might be explained due to a stronger influence of the remaining features rather than depolarization on the aerosol classification problem with the NN setup.

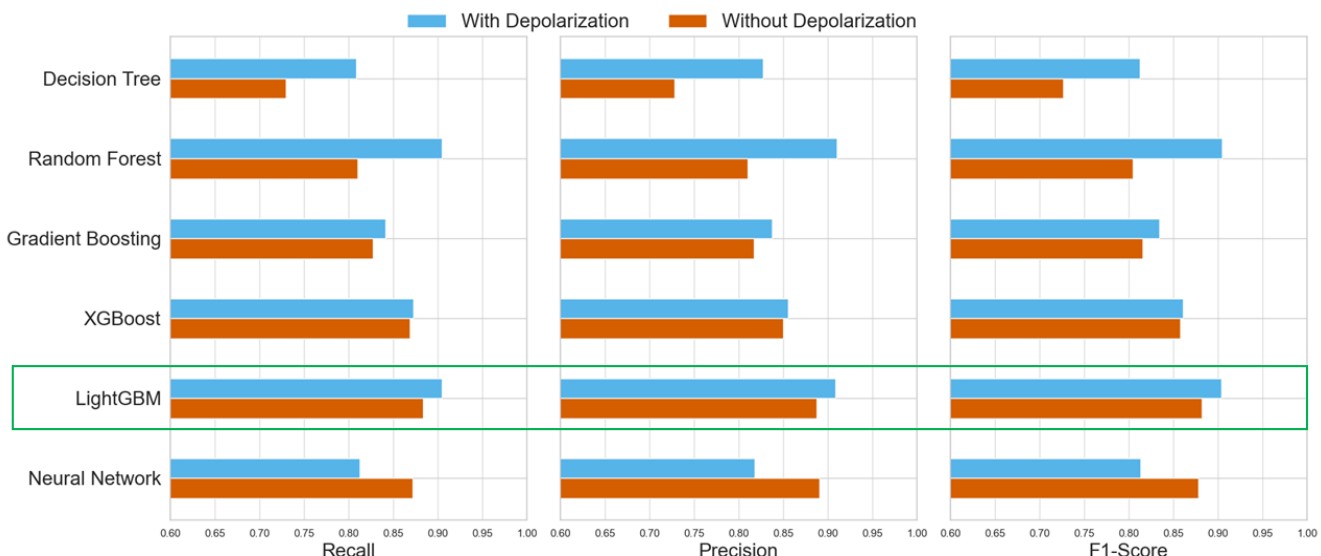

**Figure 5.** Summary of the metrics of the ML models applied to lidar data taking into account depolarization (blue) and without depolarization (orange) for all the variables in the testing dataset. The green box indicates the best metrics for both configurations.

All ML models show good performance for both configurations with all the metrics consistently above 80%, except for the Decision Tree model without depolarization, as it is the simplest model and configuration. On the one hand, the performance of the configuration without depolarization for LightGBM is above 85%, while for Random Forest it is around 80%. On the other hand, the ML models with the best performance for the depolarization configuration are Random Forest and LightGBM, achieving metrics exceeding 90%. Thus, the LightGBM model is providing the best metrics for both configurations, indicating a strong capability of this model to minimize the false positives in aerosol typing. Moreover, the recall scores above and close

to 90% for the configuration with and without depolarization, respectively, also indicate that no significant overfitting or underfitting occurred. Therefore, the ranking of models based on their performance of both configurations of the testing dataset showed that the LightGBM model achieved the best performance followed by XGBoost, Random Forest, Gradient Boosting, Neural Network and Decision Tree. Finally, these results highlight the LightGBM model robustness and generalization capacity, making it the most reliable model for the aerosol typing in this study. Thus, the LightGBM model is selected as the optimal performing algorithm in terms of recall, precision and F1-Score of the overall classification accuracy for the two configurations. This result can be explained due to the inherent nature of the LightGBM model in handling feature importance effectively, hence, selecting the most relevant variables, which aligns with the aerosol typing task.

To evaluate the computational costs of the models, we recorded the total computational time on a standard workstation equipped with a 12th Gen Intel(R) Core(TM) i7-12700H CPU and 32 GB of RAM. Simpler models such as Decision Trees and Random Forests completed training in under 20 seconds. More complex models like XGBoost and LightGBM required between 20 seconds and 1 minute. The Neural Network model required approximately 2 minutes. The Gradient Boosting model had the highest computational cost, with a training time of around 8 minutes. Despite these differences, once the models are trained, all models provide near-instantaneous predictions.

### 3.2.3 Feature importance analysis

We have applied feature importance analysis to the best performing ML model to analyze which variables are more significant for the aerosol typing classification problem. Figure 6 shows a summary of the feature importance by means of a SHAP plot for the LightGBM model. The most important features (up to twenty features) according to their mean absolute SHAP value, are shown in descending order, with the most important feature at the top. The SHAP plot shows the results for both configurations of the reference dataset.

We observe that the first feature is the most important and is the same: the average lidar ratio at 532 nm ($\overline{LR}_{532}$) of the aerosol layers. However, the second most important feature is different between the two configurations: for the configuration with depolarization the average depolarization of the layer ($\bar{\delta}_{part}$) is the most important feature, while for the configuration without depolarization is $\overline{CI}_{532,1064}$. This property is crucial as it is a strong proxy for aerosol size and type, particularly in the absence of depolarization data but also for the configuration with depolarization. The third most important feature is the altitude ($z$) for both configurations. The results highlight the relevance of the physical information of the intensive properties (e.g. CI, CR, LR, AE) in the general aerosol classification problem. It should be noted that although depolarization appears as a highly relevant feature, this does not automatically lead to improved performance for all model types, as occurred with the MLP classified (Figure 5), because the improvement depends on how each algorithm processes and learns from the available information.

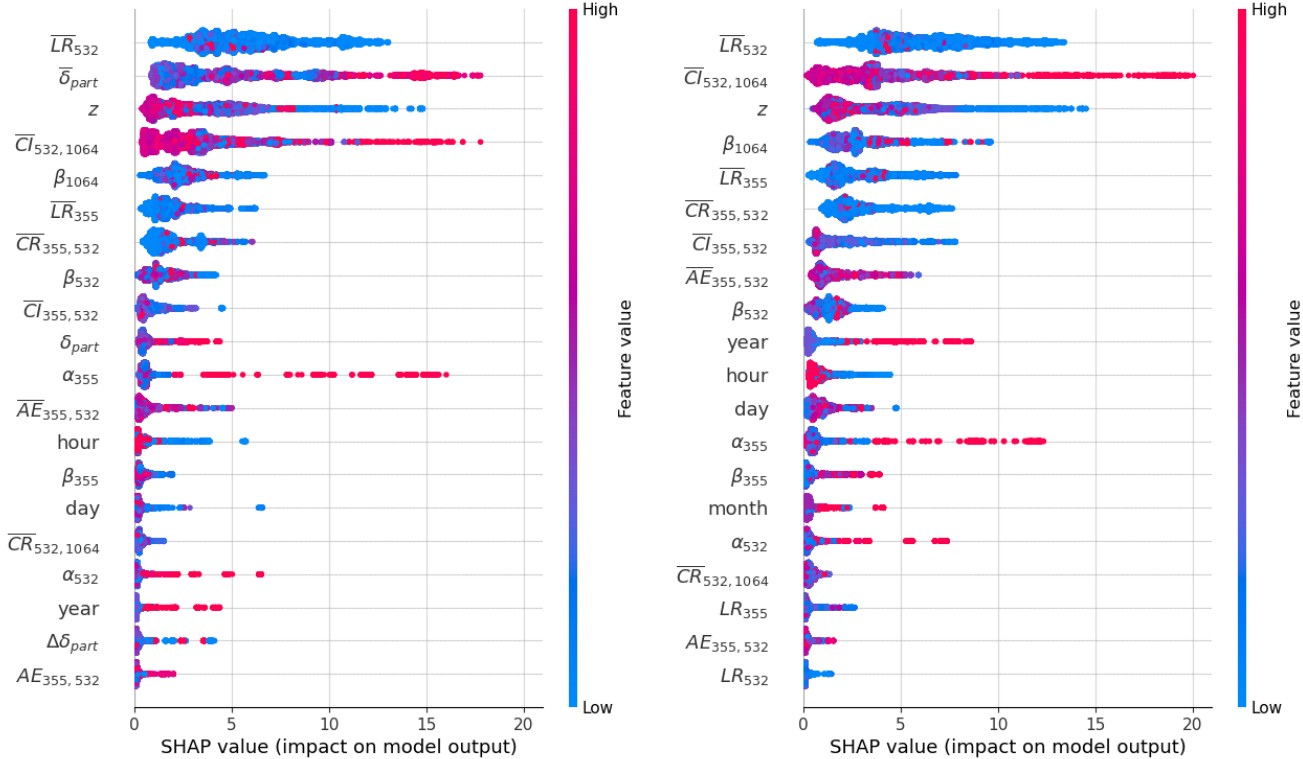

**Figure 6.** SHAP summary plots of feature importance for the LightGBM model applied to lidar data (left) taking into account depolarization and (right) without depolarization. The variables with a bar at the top indicate the average layer values, while the symbol Δ indicates the error of the variable. The rest of the variables are at high-vertical resolution.

Classical aerosol typing schemes in the literature predominantly rely on the LR and depolarization ratio at 532 nm to manually classify aerosols (e.g. Soupiona et al., 2020). In this regard, the results for the configuration with depolarization are in agreement with these traditional classification approaches, as the ML models make use of physically interpretable features, such as LR and depolarization ratio to classify aerosols.

Although averaged layer properties significantly contribute to the model performance, the SHAP plot also highlights the important role of vertically resolved features in capturing aerosol properties with precision. Features like depolarization ratio at high resolution in the configuration with depolarization, or backscatter properties at high resolution for both configurations, are crucial in the decision-making process of the ML model as they enhance the interpretability of the LightGBM model. In general, the error variables exhibit null importance for this ML model, with the exception of the depolarization error in the configuration with depolarization. This result implies that the error variables can be excluded from the model and reduce the dimensionality of the reference dataset without compromising its performance whilst optimising the feature set and potentially improving computational efficiency (del Águila et al., 2019). This is supported by the fact that LightGBM model uses Exclusive Feature Bundling (EFB), which in turn optimizes feature handling by grouping sparse and mutually exclusive

features (Ke et al., 2017). Thus, the zero importance of error variables suggests their redundancy. Furthermore, the significant importance of the depolarization error at vertical resolution for the configuration with depolarization, emphasises the unique role for this variable in the context of the aerosol typing.

### 3.2.4 Performance evaluation by aerosol type

In order to further analyse the aerosol classification capacity of the LightGBM model, we have evaluated the performance by aerosol type. Figure 7 shows the confusion matrices for the ML model under two configurations: (1) with depolarization and (2) without depolarization. This figure provides additional information for distinguishing aerosol classification, where the diagonal values represent the correct classifications, i.e., coincidence between the true and predicted labels. For the configuration with depolarization, the accuracies of the diagonal are very high, particularly for Dust (94%), Unknown (93%) and Smoke (89%) types, while the Continental Polluted (66%) and Mixed (69%) are usually misclassified with Smoke (34%) and both Smoke (12%) and Unknow (19%), respectively.

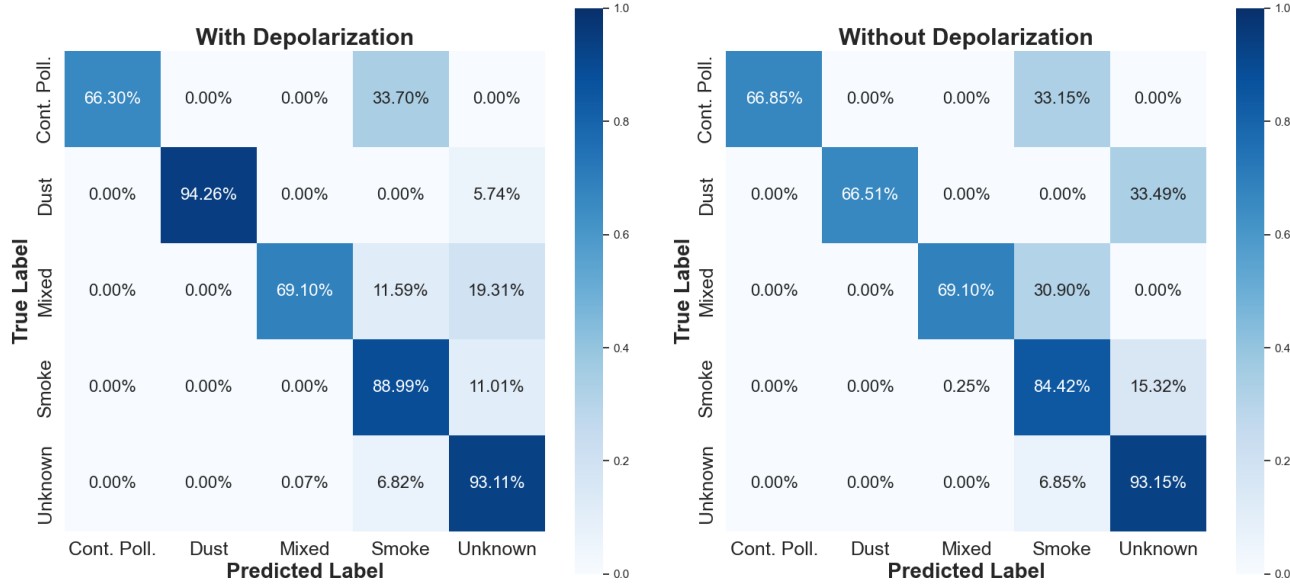

**Figure 7.** Confusion matrix of the LightGBM model for the configuration (a) with depolarization and (b) without depolarization.

For the configuration without depolarization, there is an absolute decrease of 4% in classification accuracy for Smoke and a strong decrease of 28% for Dust aerosol. This suggests that the depolarization has an important role in the classification task of those two aerosol types. However, the accuracies for Continental Polluted, Mixed and Unknown types are maintained compared to the configuration with depolarization, indicating that those types are less sensible to depolarization. Finally, we can draw the following conclusions from the figure:

- The inclusions of depolarization improve the general ability of the ML model to distinguish among aerosol types, especially for Dust and Smoke, suggesting that depolarization plays an important role in classifying these types of aerosols. This result is expected and more relevant for Dust aerosol, since dust particles are non-spherical which in turn depolarize light when measured by the lidar in a major extent (e.g. Baars et al., 2017).

- The Continental Polluted and Mixed types show similar performances (~70%) in both configurations, suggesting that these aerosol types are more challenging to predict in the classification process. This is partly due to the broad and overlapping ranges of lidar properties (e.g., LR and particle depolarization ratio) used for initial labelling, which can lead to confusion between classes such as Mixed, Continental Polluted, and Smoke. Although the Mixed type does not necessarily imply a physical dominance of Smoke, the optical properties associated with Smoke (e.g., lidar ratio from 26 to 100 sr, linear depolarization ratio from 0.001 to 0.14 in Table 1) can overlap with those of other classes, making separation difficult. Thus, a thorough expert review and the integration of additional information (e.g., backward trajectories and aerosol concentration models) were necessary to refine the final labels. Indeed, the Mixed class was less confused (around 12%) with Smoke type in the scenario with depolarization data, probably due to the smaller overlap between these classes.

- The Unknown type maintains a high accuracy of 93% for both configurations, indicating a high prediction of the unclassified aerosol types independent on the configuration.

**3.3 External validation with independent datasets: a smoke case and a Saharan dust event**

To validate the best performing ML model of this study, and evaluate its generalization capabilities, we have applied LightGBM to two independent datasets which were used for neither training nor testing the model. The first case corresponds to a less-depolarizing aerosol with predominancy of medium and long-range transported smoke that occurred on 14 September 2024 at 21:29 UTC at UGR station while, the second corresponds to a well-documented dust event discussed in Benavent-Oltra et al. (2019).

The first study case on 14 September 2024 presented several layers that were annotated as long- and medium-range transported smoke from Canada and North of Portugal, respectively, thanks to the ancillary information of model backward trajectories and satellite global fire map products (not shown here). This event was captured by a different multiwavelength Raman lidar, ALHAMBRA, that also provided, among other products, $\beta_{par}$ at 355, 532 and 1064 nm, $\alpha_{part}$ at 355 and 532 nm. The retrieved profiles and their derived products are shown in Fig. 8. The predictions with LightGBM of the first aerosol layer correspond to the Continental Polluted aerosol while for the second to fourth layers, the predictions correspond to Smoke aerosol. Thus, the predictions are coincident with the actual aerosol type, as previously classified by experts following the criteria of Section 2.1. Hence, the correct predictions of this independent dataset prove the generalization of the ML model due and accurate prediction for another instrument with different spatial resolution (3.5 m) and for low-depolarizing aerosols such as Smoke and Continental Polluted aerosol.

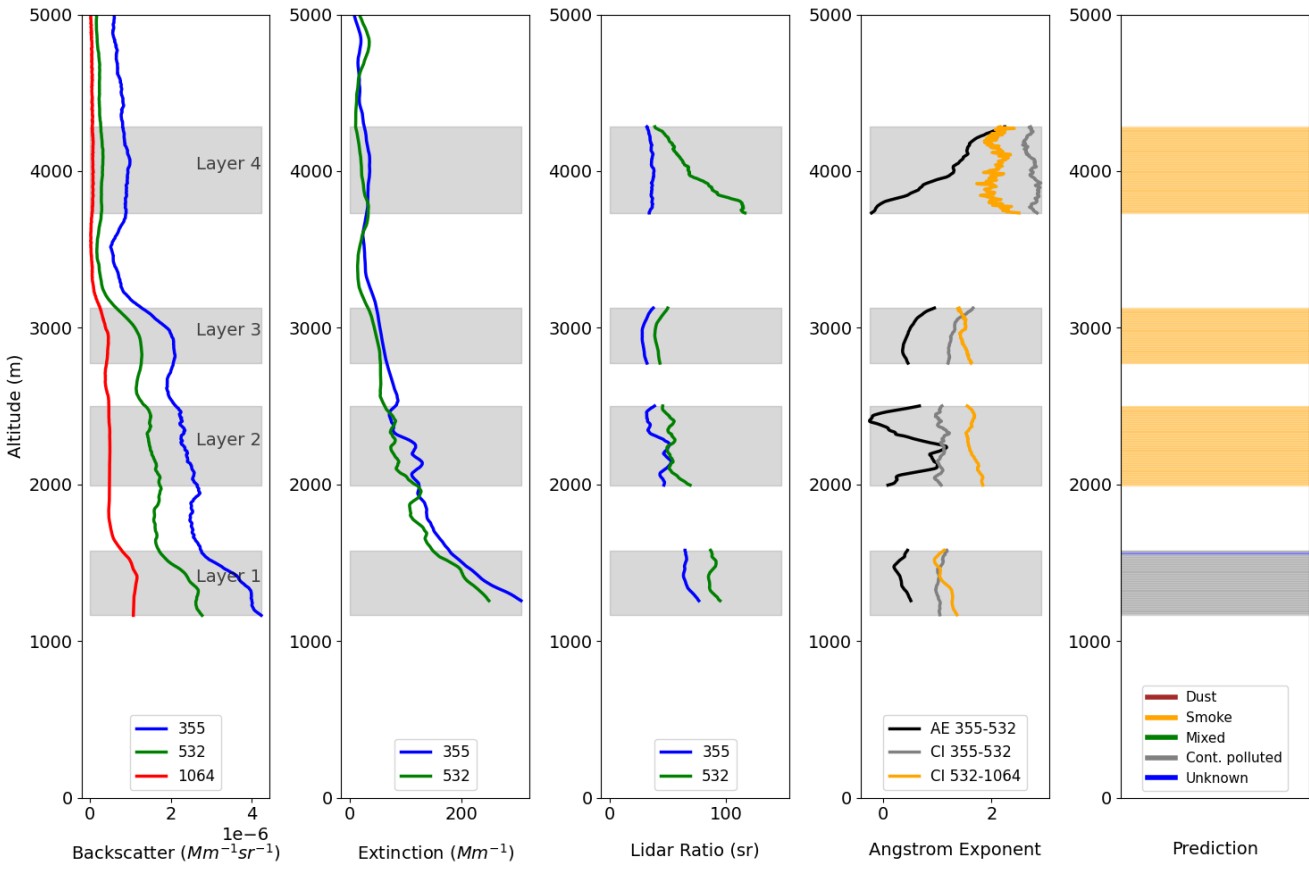

**Figure 8.** Vertical profile of aerosol optical properties with layers and classification for Granada on 14 September 2024 at 21:29 UTC obtained from EARLINET. (a) Backscatter coefficient profile at 355, 532 and 1064 nm. (b) Extinction coefficient profiles at 355 and 532 nm. (c) Lidar ratio at 532 nm. (d) Angstrom exponent and Color Ratio. (e) Aerosol type prediction with trained LightGBM model. The lidar data from this case corresponds to the ALHAMBRA lidar.

The second study case is an intense Saharan dust event occurred during a field campaign called SLOPE I, from 18 to 21 July 2016. During this campaign, the MULHACEN multiwavelength lidar measured at three wavelengths (355, 532 and 1064 nm) with predominant Dust and Mixed aerosol types, and containing a number of height-resolved values classified as Dust of 1527 and 523 for Mixed. Moreover, the independent dataset comprises 26 aerosol layers, providing a comprehensive case for testing the performance of our classification model in unseen data, specially under challenging atmospheric conditions. Saharan dust events are frequent on the Iberian Peninsula and several efforts in previous studies have been made to correctly identifying dust aerosol (e.g. Córdoba-Jabonero et al., 2018; López-Cayuela et al., 2023).

The independent dust event dataset was manually labelled following the same criteria described in Section 2.1., in order to test the accuracy of the ML model. Thus, we have validated the LightGBM model for the configuration with depolarization with

the independent dataset of the Saharan dust event for the same lidar instrument. The accuracy results demonstrate good performance in classifying the predominant aerosol type, by correctly classifying 82% of Dust aerosol instances, and F1-score of 90%. However, there are some misclassifications between Dust and Unknown aerosols, with 18% of Dust samples classified as Unknown and the Mixed samples are mainly confused with Unknown aerosol and in a minor extent with Continental polluted aerosol, indicating some overlap in their features. Finally, the Unknown classification is of 100%. Overall, the model struggles more with Mixed aerosols, but the accuracy is very high for Dust classification, which is the major aerosol component during the Saharan dust event (78% of the aerosol types).

The profile evolution of the dust event has been also analyzed to assess the model's ability to capture the spatial and temporal variability of the aerosol layers. Figure 9 shows the vertical profiles of backscatter, extinction, LR and AE during the event, on 19 July 2016 at 22 UTC, along with the predicted aerosol types for the five detected layers. Comparing the predictions with the reference dataset, Layer 1 was classified as Mixed but it is predicted by the LightGBM model as Unknown (blue). This discrepancy with the first layer occurs for two other profiles, which could indicate that the model struggles to classify aerosols on the bottom layer, potentially due to the complex mixture of aerosol types or limited feature representation for this particular case. For Layers 2, 3 and 4, both the reference and the ML model classify them as Dust, showing strong agreement in identifying this layer's predominant aerosol type. Finally, Layer 5 was classified as Unknown in the reference dataset and is also predicted as Unknown by the model, which suggests that the ML model encounters occasional uncertainty in distinguishing the aerosol types, particularly when several intensive properties lack of information, leading to its classification as Unknown.

In conclusion, the LightGBM model achieves top accuracy in predicting intermediate layers, whereas the bottom and top layers are usually misclassified (35% of the total amount of layers) probably caused due to lack of information about the intensive properties of those layers. Additionally, there might be overlapping features or presence of secondary aerosol components that contribute to confusion on the bottom and top layers.

To further evaluate the performance of the LightGBM ML approach, we compared it with the automatic algorithm for aerosol classification called NATALI (Nicolae et al., 2018), which is also applied to EARLINET data. We executed NATALI version 1.4.1. (Nicolae et al., 2016) on the independent dataset of the Saharan dust event from Section 3.3 for classifying aerosol types on the detected layers. It is worth mentioning that the detected layers are the same for both models, so that the two models are under comparable conditions. Depolarization information was unavailable in the EARLINET database during that period; however, depolarization data was provided by the research group for the same event. As a result, NATALI algorithm provides a "low-resolution typing", able to identify 5 predominant aerosol types (Nicolae et al., 2018). When evaluating LightGBM without depolarization data, its performance was comparable to that of NATALI. However, when depolarization data was included, the performance of LightGBM improved significantly, correctly identifying the predominant aerosol type in 65% of the layers, compared to 23% for NATALI under the same dataset. This highlights the critical role of depolarization in differentiating challenging aerosol types, such as dust and smoke.

The differences in classifying the aerosol types between the two models can be explained by the different aerosol type definition between the two automatic typing models (Voudouri et al., 2019). NATALI uses synthetic data to train the neural networks and is conservative in classification, often leaving layers unclassified due to high errors or missing intensive parameters. On the other hand, the approach presented in this work uses real data at high-resolution and allows for classification even in cases where certain variables (such as intensive parameters) cannot be calculated or have high uncertainties.

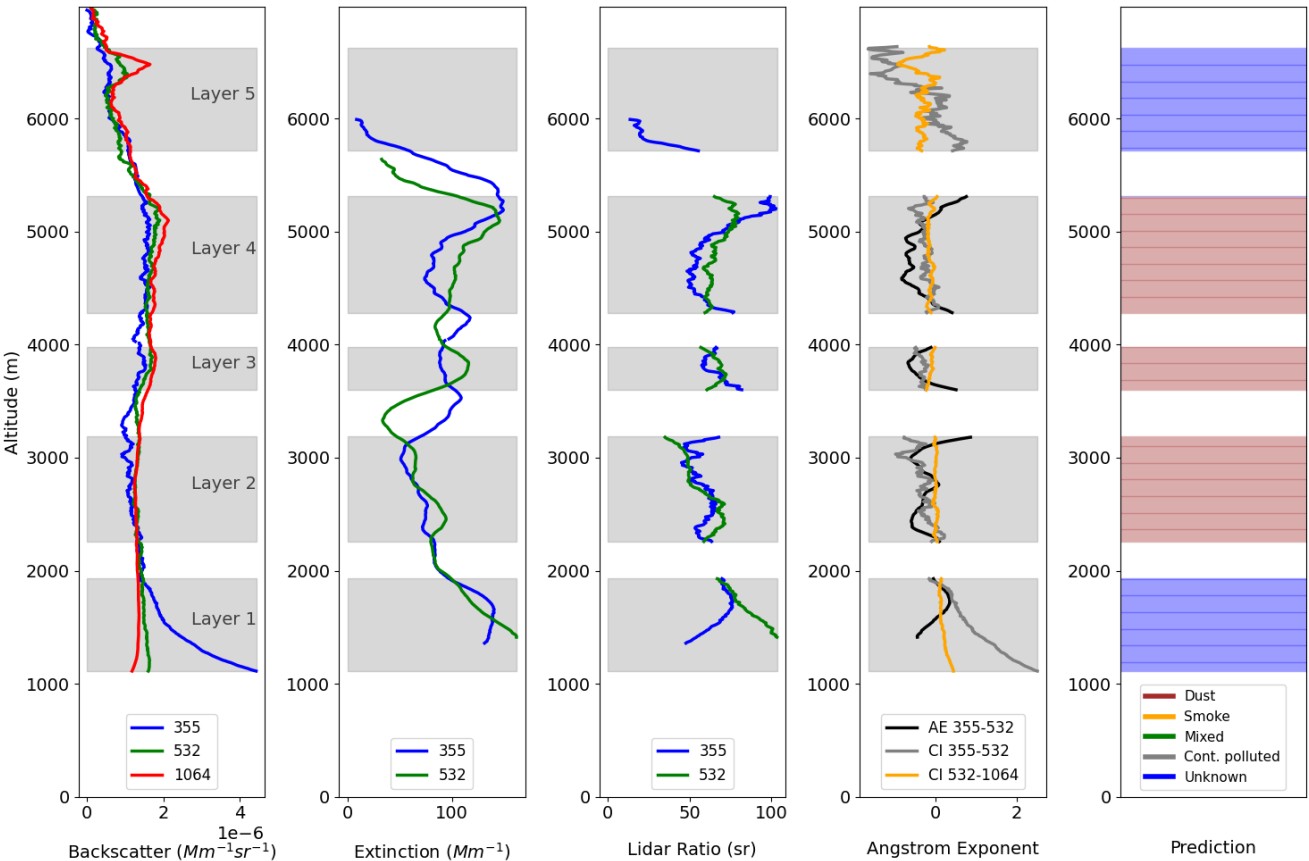

**Figure 9.** Similar to Figure 8 but for the date 19 July 2016 at 22:00 UTC obtained from EARLINET. The lidar data from this case corresponds to the MULHACEN lidar.

## 4 Discussion

After the evaluation of six ML models, the best-performing ML model was LightGBM, using two data configurations: with depolarization and without depolarization. Given its strong performance in classifying aerosol types with both configurations, this model could serve as a benchmark for future aerosol classification efforts, knowing which ML model is more suitable for this purpose. Its ability to generalize across different atmospheric conditions, while maintaining high accuracy, makes it a valuable starting point for comparisons in future studies or implementations across various lidar networks.

Our study introduces several key innovations in the field of aerosol classification of lidar data using ML:

1. Design and preparation of a reference dataset for ML applications: we have designed and created a reference dataset from scratch of multiwavelength lidar data, including extensive and intensive properties with their corresponding errors at high vertical resolution. In addition, we have manually labelled the detected layers. This information is crucial for having a reference dataset and for applying different supervised ML models.

2. High-resolution aerosol typing: this study represents the first instance of aerosol typing conducted using high-resolution EARLINET data, rather than the conventional approach of using averaged aerosol layer values. While high-resolution data inherently introduces higher uncertainty or noise, the results show good performance and the ML models manage to accurately classifying aerosol types at high resolution. In this regard, the Unknown class plays a crucial role in accounting for cases when insufficient information prevents a definitive classification.

3. Validation of satellite products: The approach of this work will contribute to validate aerosol typing of satellite remote sensing missions like EarthCARE (Wehr et al., 2023) at high resolution.

4. Applicability to other lidar stations: Although our model is currently trained for the UGR station, the methodology can be extended to other EARLINET stations and serve as benchmark dataset. By training the model with additional aerosol types that are prevalent in other locations, the model could predict aerosol types in various regions with different compositions. For instance, incorporating some representative EARLINET stations with diverse aerosol types, such as marine or continental polluted aerosols, could enhance the model performance in other parts of the globe.

5. Computation of extinction-to-mass coefficients: By identifying the aerosol type for each layer using the ML approach, different extinction-to-mass conversion factors tailored to specific aerosol types can be applied, thereby reducing uncertainties in the estimation of particle mass concentration ($PM_{10}$) [$\mu g/m^3$].

Our model is currently trained with data from the UGR station, which means that it is primarily designed for the specific aerosol types present in this region. However, this also presents an opportunity: the methodology developed in this study can be extended to similar instruments within the same station (Alados-Arboledas & Guerrero-Rascado, 2024), to other lidar stations within the EARLINET network or similar lidar databases like DeliAn (Floutsi et al., 2023), enabling the creation of region-specific labelled datasets for aerosol classification. This flexibility allows for adaptation to other geographical locations, provided that the aerosol types for those regions are included in the training set. Future research should focus on expanding the dataset to include more aerosol types. Furthermore, extending the model's application to real-time aerosol classification across multiple lidar stations could significantly enhance the operational capability of lidar networks like EARLINET.

## 5. Conclusions

This work highlights the effectiveness and versatility of machine learning (ML) models for aerosol typing using high-resolution EARLINET data. Among the tested models, LightGBM demonstrated superior performance, achieving up to 90% accuracy after hyperparameter optimization, outperforming existing approaches. In addition, it has been found that the linear particle

depolarization ratio is as a key feature for classifying aerosol types, particularly dust, but also show that the model remains

robust when such features are unavailable, achieving 83% accuracy in external validation with fewer features or high uncertainties.

The LightGBM model was successfully validated against two independent datasets representing a Saharan dust event and less-depolarizing aerosol with predominancy of medium and long-range transported smoke, demonstrating consistent performance across layers and aerosol types. This work introduces several innovations, including the development of a high-resolution

reference dataset, the prediction of "Unknown" classes and its implications with real measurements, and the potential for validating satellite aerosol products at high resolution. The use of vertically-resolved data plays a key role in accurately predicting aerosol types at high resolution, as supported by the feature importance analysis. Furthermore, the methodology can be extended to other lidar stations within EARLINET, enabling the inclusion of region-specific aerosol types and enhancing its applicability across diverse geographical areas.

Future research should focus on expanding the dataset to include more aerosol types and exploring unsupervised ML approaches. The implementation of real-time aerosol classification across lidar stations could significantly enhance the operational capabilities of lidar networks like EARLINET, contributing to a better understanding of atmospheric composition and its impact on the climate.

### Data availability

The files of the lidar MULHACEN at UGR (Spain) station are available at EARLINET archive (https://www.earlinet.org/index.php?id=earlinet_homepage). The reference dataset used in this study, including processed layers and manual classification is openly available at Zenodo [DOI: 10.5281/zenodo.16925786].

### Author contributions

AdA and LAA had the idea for the analysis. AdA analyzed the data, conducted the original research and wrote the manuscript
draft. AdA and POA annotated the dataset. AdA and ST contributed to the design of the ML methodology. AdA, POA and ST contributed to the formal analysis. POA, ST, JBA, SFC and LAA reviewed and edited the manuscript.

### Competing interests

The authors declare that they have no conflict of interest.

### Acknowledgements

This research is part of the Spanish national projects PID2023-151817OA-I00, PID2020-120015RB-I00 and PID2022-142708NA-I00, funded by MICIU/AEI/10.13039/501100011033 and by the "European Union NextGenerationEU/PRTR". It is also supported by ACTRIS-IMP grant agreement No 871115, ACTRIS-España (RED2022-134824-E), Scientific Unit of Excellence: Earth System (UCE-PP2017-02), and by University of Granada Plan Propio through Excellence Research Unit Earth Science and Singular Laboratory AGORA (LS2022-1) programs. AdA is part of Juan de la Cierva programme through

grant JDC2022-048231-I funded by MICIU/AEI/10.13039/501100011033 and by European Union "NextGenerationEU"/PRTR. POA is funded by European project ATMO ACCESS - Solutions for Sustainable Access to Atmospheric Research Facilities (Ref. 101008004). SFC received funding from the Spanish Ministry of Research and Innovation (Agencia Estatal de Investigación), grant PRE2021-098351 (co-funded by the European Social Fund Plus).

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
