# Peer review of "Aerosol type classification with machine learning techniques applied to multiwavelength lidar data from EARLINET"

_EGUsphere, 2025_

## Referee Comment (RC1)

**EGUsphere-2025-269: "Aerosol type classification with machine learning techniques appliedto multiwavelength lidar data from EARLINET", by del Águila et al.**

This paper presents a novel methodology for automated aerosol classification based on Machine Learning (ML) methods applied to lidar measurements with respect to other aerosol typing procedures. Indeed, aerosol typing, especially with high vertical resolution, is a crucial issue for a better understanding of atmospheric composition and its impact on the climate. Therefore, the outcomes of this work are rather relevant and hence it deserves to be published. However, a Major revision should be performed before it is accepted for publication.

General comments:
1) Main findings of this work are critically based on a first aerosol classification manually performed using the EARLINET Granada lidar database (aerosol optical properties and their derived intensive parameters, as shown in Eqs. 1-4). However, no information is introduced in the work about the criteria used for that purpose. This is crucial as the performance of the ML methods applied in aerosol classification is compared against that manual typing. It should be included and clarified (an additional table could help).
2) The reference dataset in divided in 80% for training and 20% for testing of the ML methods. Could the results be affected if those percentages are modified? Please, provide an explanation.
3) An external validation of the best ML method is performed with an independent dataset just for a dust case. Why it is not validated for another type of aerosols, for instance, less-depolarizing aerosols (e.g. smoke or continental pollution),and thus to be more confident in the ML method performance? This would improve the work.
4) What is the difference of using either height-resolved data or conventional average aerosol layer values (i.e. the same value at every height-level of the layer)? This can trigger differences in the results. Please, clarify.

Specific comments:
1) Sect. 2.1:
   a) Once the aerosol layers are identified, the paper indicates that the database includes two representations of the intensive properties (page 4, line 111): height-resolved data and layer-averaged data. The latter averaged value is then assigned to every height-level of each particular identified layer to maintain the lidar resolution. If a single type of aerosol is assigned to each aerosol layer (page 4, line 117), this procedure just serves to increase the same averaged value, which is also associated to a given aerosol type. This should be explained in more detail.
   b) Also, specify what of the two data representations is used as the reference dataset for comparison with the outputs of each ML method (page 4, lines 115-117).
   c) The vertical resolution of the lidar profiles used in the reference database for the UGR station is indicated but not their time resolution (page 4, lines 92-94). This should be included.
2) Sect. 2.6: Please, provide more information on:
   a) Table 1: Why did you applied those ranges of values for each hyperparameter?
   b) Page 8, lines 212-214: Which are the weights to each class applied?
3) Sect. 3.1: Figures 3 and 4, and Table 2 (pages 10-11): They can be affected depending on the manual aerosol classification performed to differentiate the 'Unknown' type from the rest of aerosols, and between each aerosol class. Please, clarify.
4) Sect. 3.2:
   a) Page 12: Table 3 could be merged with Table 1.

b) Page 12, lines 281-282: It is not clear if the 416 layers examined in this work correspond to the reference dataset or the testing dataset. Please, clarify.

c) Pages 12-13, lines 284-285: In the statement: 'In general, the ML models that incorporate depolarization data demonstrate significantly higher performance compared to those without depolarization', it should be added 'except for the neural network', as shown in Figure 5.

d) Page 13, lines 288-289: In the sentence: 'This might be explained due to a stronger influence of other features rather than depolarization on the aerosol classification problem with the NN setup'. Explain what do you mean by 'other features', please.

e) Page 16, lines 362-365: In the statement, 'The Continental Polluted and Mixed types show comparable performances near ~70% in both configurations, indicating that these aerosol types are more difficult to be predicted in the classification process. They are also confused with Smoke in a ~30%, which makes sense due to nature of the Mixed aerosol type, which can be composed of several types of aerosols, being Smoke one of the major contributors.', why Smoke is one of the major contributors in the Mixed type? Maybe that is related to the way the Mixed class was identified? Please, clarify it.

5) Sect. 3.3:
a) Page 17, line 397: Please, provide more detail for '… particularly when several intensive properties are missing, …'. Why?

b) Pages 18-19, lines 413-418: Please, check this paragraph, there is an inconsistency. If there are not available depolarization information in the EARLINET database during the Saharan dust period, how the depolarization data could be included for the evaluation of the LightGBM performance in this configuration? Please, clarify.

6) Discussion Sect.:
a) Page 19, lines 431-434: Although it was mentioned in this section, the manual aerosol classification is not described in the work, and it can affect the reference dataset used. See General Comment #1.

b) Page 19, lines 435-439: See General Comment #4.

Other minor comments:
1) Page 5, line 121: What do you mean with 'corresponding height of each layer'? Please, specify.
2) Page 5, line 128: What is the meaning of 'row' in this sentence?
3) Page 5, line 135: Why do you use the median value for the missing values? Explain.
4) Page 5, line 136: If each profile is denoted by $j$, does it have sense $j$=0? Also, define $\Delta z$.
5) Page 6, lines 170-172: If $k$ denotes the variables, $k$ = 0 should be avoided, shouldn't it? Also, the term 'the predicted class for each layer and height' is correct? Either layer or height?
6) Page 7, line 198: Please define 'GroupKFold' (and include reference).
7) Page 7, line 198: Please define 'GroupShuffleSplit' (and include reference).
8) Page 10, Figure 3's caption: 'ratio' is missing at the end of 'particle linear depolarization'.
9) Page 3, Figure 5's caption: Should it be 'in the testing dataset' instead of 'in the reference dataset'?
10) Page 14, Figure 6: The title in the colour bar may be 'Feature importance' instead of 'Feature value'?
11) Page 17, line 371: Check: ' … which was used for neither training nor testing the model, …'.
12) Page 18, Figure 8: Check the x-axis: the title of the first panel, and also the scale in the second panel (it should be multiplied by $10^6$).
13) Pages 21-28: Check the reference list as wrong and missing details are found.

---

## Author Response (AR1)

**Response to Referee #1 – del Águila et al.:**

First of all, we would like to thank the reviewer for the positive comments that definitely have helped to improve the manuscript. In the following, a point-by-point response to the reviewer's comments is included below. Reviewer's comments are written in **bold** while the answers to each comment are written in regular format. Changes in the manuscript are noted between quotation marks ("") and underlined, and are referred to the corresponding line in the revised version of the manuscript.

**Reviewer #1: This paper presents a novel methodology for automated aerosol classification based on Machine Learning (ML) methods applied to lidar measurements with respect to other aerosol typing procedures. Indeed, aerosol typing, especially with high vertical resolution, is a crucial issue for a better understanding of atmospheric composition and its impact on the climate. Therefore, the outcomes of this work are rather relevant and hence it deserves to be published. However, a Major revision should be performed before it is accepted for publication.**

General comments:

**1) Main findings of this work are critically based on a first aerosol classification manually performed using the EARLINET Granada lidar database (aerosol optical properties and their derived intensive parameters, as shown in Eqs. 1-4). However, no information is introduced in the work about the criteria used for that purpose. This is crucial as the performance of the ML methods applied in aerosol classification is compared against that manual typing. It should be included and clarified (an additional table could help).**

We thank the referee for this comment. We agree with the reviewer that the information included in the manuscript about the manual classification was not enough. There was some information in lines 115-122 on the original manuscript. Also, as suggested by the reviewer, we have included a table with the ranges employed as first approach for the manual classification. Thus, we have further explained the manual typing criteria by adding lines 120-122 and 125-128 (underlined) of the revised version of the manuscript:

"Based on the literature (Groß et al., 2013; 2014; Navas-Guzman et al., 2013; Illingworth et al., 2015; Soupiona et al., 2020), we assigned a single type of aerosol to each aerosol layer at average resolution according to certain ranges of the calculated average intensive properties, as described in Table 1. These properties range criteria were a first attempt for aerosol classification. This initial approach was followed by a thorough review of each vertical profile and its optical properties by experts. Corrections were made to the aerosol type in cases of misclassification identified during the first attempt. Finally, to ensure accurate classification, we employed ancillary information to verify the aerosol type assigned to each layer. Therefore, further analysis was carried out by running HYSPLIT backtrajectories (Stein et al., 2015) and NAAPS model (Lynch et al., 2016) to support the labelling. For the cases where the aerosol type was not clear enough, we computed the backtrajectories with the HYSPLIT model, taking as the starting point of the altitude for the backtrajectory analysis the one of each identified aerosol layer. Thus, for uncertain aerosol layer types, we ran the model for 5 days in advance to trace the air mass origin for the altitude corresponding to each layer. In addition, we assessed temporal consistency by verifying that the aerosol types between layers remained consistent from subsequent profiles."

The following table is included in Section 2.1 (lines 130-133) of the revised manuscript:

**Table 1.** Indicative ranges of lidar properties for manual aerosol typing at UGR station, adapted for this study and based on typical ranges in the literature, were used as an initial reference for the manual labelling process. This first attempt for aerosol classification was followed by a thorough review of each vertical profile and corrections were made when necessary to ensure accurate classification. Lidar ratios (LR) are expressed in steradians (sr).

| Aerosol type | Properties range criteria |
|---|---|
| Clean Cont. | $22 \leq LR_{355} \leq 36$ & $0.02 \leq \delta_{part} \leq 0.06$ |
| Volcanic | $30 \leq LR_{532} \leq 60$ & $0.33 \leq \delta_{part} \leq 0.46$ |
| Smoke | $26 \leq LR_{532} \leq 100$ & $0.001 \leq \delta_{part} \leq 0.14$ |
| Dust | $32 \leq LR_{532} \leq 71$ & $0.1 \leq \delta_{part} \leq 0.32$ |
| Mixed | $32 \leq LR_{532} \leq 71$ & $0.1 \leq \delta_{part} \leq 0.2$ |
| Cont. Polluted | $42 \leq LR_{532} \leq 81$ & $0.025 \leq \delta_{part} \leq 0.07$ & $1.7 \leq CR_{355,1064} \leq 2.7$ & $0.4 \leq AE_{355,532} \leq 1.6$ |
| Unknown | Else |

**2) The reference dataset in divided in 80% for training and 20% for testing of the ML methods. Could the results be affected if those percentages are modified? Please, provide an explanation.**

The partition of the dataset into 80% for training and 20% for testing is a widely used and well-established practice in machine learning (ML). This standard division is commonly adopted because it usually provides a good balance between the amount of data available for training and an adequately sized test set to assess the model performance (Kohavi, 1995).

The size of the training set influences the model's ability to learn the patterns and generalize to new samples that were not seen during the training process. On the one hand, if the training set percentage is increased, the model may have more data to learn from, which could improve its generalization capability. However, if the test set percentage is reduced too much, the evaluation of the model's performance could become unreliable. Thus, a smaller test set could fail to represent the full variability of the data, leading to misleading performance metrics (Zhou, 2017). On the other hand, if the training set is reduced, the model might not have enough information to learn the necessary patterns, which could negatively affect its performance. In addition, an inadequate training data might result in overfitting, where the model performs well on the training set but poorly on unseen data (Bishop, 2006). Thus, the 80/20 split is chosen to find a balance between having enough data to train the model effectively and ensuring that the model is evaluated on a sufficiently diverse test set.

In order to clarify this division of the dataset, we have included in the manuscript the following explanation in lines 221-224 of the revised version of the manuscript:

"The 80/20 split is a commonly adopted practice in ML, as it provides a good balance between the amount of data available for training and a sufficiently large test set for reliable performance evaluation (Kohavi, 1995). This proportion is usually considered to offer an optimal balance, ensuring that the model is evaluated on a sufficiently diverse test set and maintains the integrity of the evaluation process (Zhou, 2017)."

References included in the reference list of the revised manuscript:

- Kohavi, R.: A study of cross-validation and bootstrap for accuracy estimation and model selection, *Proc. 14th Int. Joint Conf. on Artificial Intelligence (IJCAI)*, Montréal, Canada, 1137–1143, 1995.

- Zhou, Z.-H.: A brief introduction to weakly supervised learning, *National Science Review*, 5(1), 44–53, https://doi.org/10.1093/nsr/nwx106, 2018.

Reference included in this response but not included in the manuscript:

- Bishop, C. M.: *Pattern Recognition and Machine Learning (Information Science and Statistics)*, Springer-Verlag, Berlin, Heidelberg, 2006. https://doi.org/10.5555/1162264.

**3) An external validation of the best ML method is performed with an independent dataset just for a dust case. Why it is not validated for another type of aerosols, for instance, less-depolarizing aerosols (e.g. smoke or continental pollution),and thus to be more confident in the ML method performance? This would improve the work.**

We agree with the reviewer that adding an additional case would improve the overall performance of the selected ML method and the manuscript. Thus, we have included a less-depolarizing aerosol case that contains Continental Polluted and Smoke aerosol types, from an independent dataset and another multiwavelength lidar at UGR station: ALHAMBRA. This smoke case, with aerosol layers detected thanks to the temporal evolution of the range-corrected signal (Figure R1) occurred on 14/09/2024 and the annotation was supported by ancillary information from the backward trajectories from HYSPLIT (Figure R2) and the FIRMS Fire alert system (Figure R3). With this supporting information we could confirm that the lidar profile presented some layers that were medium-range transported from fires in the North of Portugal and some others long-range transported from Canadian fires. In order to improve the work, as suggested by the reviewer, we have included in Section 3.3 a new figure, similar to Figure 8 from the original manuscript, but with the smoke case, as explained below.

[Figure]

*Figure R1. Temporal evolution on the range corrected signal (RCS) at 532 nm for the day 14/09/2024. The selected case is one hour profile from EARLINET at 21:29 UTC which is represented by the vertical white dashed lines.*

[Figure]

*Figure R2. Backward trajectories from HYSPLITT for the day 14/09/2024 at 21:30 UTC at different heights.*

*Figure R3. Satellite imagery from FIRMS showing active fire hotspots in Canada on 02/09/2024 (top) and in North of Portugal on 13/09/2024 (bottom), coinciding with the time when the backward trajectories arriving to UGR station passed over those locations.*

The title of the section 3.3, new comments and a new figure have been included into the revised version of the manuscript. First, we have modified the title of the section in line 414 as:

"3.3 External validation with independent datasets: a smoke case and a Saharan dust event"

Then, we have included the following text and figure in Section 3.3 in lines 416-435 of the revised version of the manuscript:

"To validate the best performing ML model of this study, and evaluate its generalization capabilities, we have applied LightGBM to two independent datasets which were used for neither training nor testing the model. The first case corresponds to a less-depolarizing aerosol with predominancy of medium and long-range transported smoke that occurred on 14 September 2024 at 21:29 UTC at UGR station while, the second corresponds to a well-documented dust event discussed in Benavent-Oltra et al. (2019).
The first study case on 14 September 2024 presented several layers that were annotated as long- and medium-range transported smoke from Canada and North of Portugal, respectively, thanks to the ancillary information of model backward trajectories and satellite global fire map products (not shown here). This event was captured by a different multiwavelength Raman lidar, ALHAMBRA, that also provided, among other products, $\beta_{par}$ at 355, 532 and 1064 nm, $\alpha_{part}$ at 355 and 532 nm. The retrieved profiles and their derived products are shown in Fig. 8. The predictions with LightGBM of the first aerosol layer correspond to the Continental Polluted aerosol while for the second to fourth layers, the predictions correspond to Smoke aerosol. Thus, the predictions are coincident with the actual aerosol type, as previously classified by experts following the criteria of Section 2.2. Hence, the correct predictions of this independent dataset prove the generalization of the ML model due and accurate prediction for another instrument with different spatial resolution (3.5 m) and for low-depolarizing aerosols such as Smoke and Continental Polluted aerosol.
The second study case is an intense Saharan dust event …"

[Figure]

**Figure 8.** Vertical profile of aerosol optical properties with layers and classification for Granada on 14 September 2024 at 21:29 UTC obtained from EARLINET. (a) Backscatter coefficient profile

at 355, 532 and 1064 nm. (b) Extinction coefficient profiles at 355 and 532 nm. (c) Lidar ratio at 532 nm. (d) Angstrom exponent and Color Ratio. (e) Aerosol type prediction with trained LightGBM model. The lidar data from this case corresponds to the ALHAMBRA lidar."

In addition, in order to avoid repetition in the captions between Figures 8 and 9, we have modified the caption of Figure 9 in lines 483-484 from the revised version of the manuscript:

**"Figure 9.** Similar to Figure 8 but for the date 19 July 2016 at 22:00 UTC obtained from EARLINET. The lidar data from this case corresponds to the MULHACEN lidar."

We have also mentioned this new case in the Conclusion Section in lines 527-528 of the revised manuscript as:

"The LightGBM model was successfully validated against two independent datasets representing a Saharan dust event and less-depolarizing aerosol with predominancy of medium and long-range transported smoke, demonstrating consistent performance across layers and aerosol types."

Finally, we have mentioned it also in the Abstract (lines 19-20) of the revised manuscript:

"Validation against independent datasets, including a smoke case and a Saharan dust event, confirmed robust classification under real and extreme conditions."

**4) What is the difference of using either height-resolved data or conventional average aerosol layer values (i.e. the same value at every height-level of the layer)? This can trigger differences in the results. Please, clarify.**

The main difference between using height-resolved data and conventional average aerosol layer values lies in the granularity of the information. Height-resolved data provides a detailed, layer-specific profile of variables, meaning that for each altitude (row), there is a corresponding value for each variable (column), such as backscatter, extinction, aerosol type, etc. This results in a more comprehensive and more informed dataset, where the relationship between aerosol properties and altitude can be captured at multiple levels.

In contrast, conventional average aerosol layer values represent a single averaged value for each variable over the entire layer. This approach simplifies the dataset by reducing the complexity, as only one value is provided per variable for the entire layer. While this might make the dataset more compact, it lacks the spatial resolution that height-resolved data provides.

By using height-resolved data, we obtain a matrix of size $n$ x $m$, where $n$ is the number of altitude levels within a layer and $m$ is the number of variables in the dataset. This allows for the inclusion of vertical profiles and the ability to observe vertical gradients, which could be crucial for understanding aerosol behavior more accurately, especially in cases where aerosol properties change significantly with altitude. This additional level of detail is especially beneficial for ML models, as they can take advantage of a richer, more complex dataset to better capture patterns and relationships within the data. The higher resolution allows the model to make more informed predictions, as it has access to a larger set of features and can identify finer variations that might be missed when using average values. In fact, in Figure 8 of the revised version of the manuscript, at the top of the first layer, the predictions provide "Unknown" class, due to the prediction at high-vertical resolution, meaning that this approach provides a more detailed prediction.

On the other hand, using average aerosol layer values treats the entire layer as a uniform entity, potentially losing vertical information. This could lead to a loss of important variations in the aerosol properties at different altitudes, which may affect the accuracy of the results, particularly in situations where vertical stratification is significant. In ML training, the reduced complexity of

using average values may limit the model's ability to generalize effectively, as the model may not fully learn the vertical dependencies between features.

In summary, height-resolved data provides a more detailed and dynamic representation of aerosol properties, which is not only crucial for a better understanding of the vertical distribution of aerosols but also offers significant advantages for ML training. The ability to capture more granular features can lead to more accurate and robust ML models, especially in cases where vertical gradients play an important role. Conventional average values, while simpler, may not capture important vertical variations, potentially leading to less accurate results and weaker model performance in ML applications.

To further clarify this point, we have included the following sentence in lines 112-116 of the revised version of the manuscript:

"The use of height-resolved data provides a detailed representation of the vertical distribution of aerosol properties, while the layer-averaged values, repeated across all height levels within each layer, provide a consistent profile representative of the overall aerosol type. This combination allows ML models to learn from both fine-scale variability and averaged layer characteristics, which is particularly beneficial for improving prediction accuracy and capturing more complex aerosol variations within layers."

**Specific comments:**

**1) Sect. 2.1:**

**a) Once the aerosol layers are identified, the paper indicates that the database includes two representations of the intensive properties (page 4, line 111): height-resolved data and layer-averaged data. The latter averaged value is then assigned to every height-level of each particular identified layer to maintain the lidar resolution. If a single type of aerosol is assigned to each aerosol layer (page 4, line 117), this procedure just serves to increase the same averaged value, which is also associated to a given aerosol type. This should be explained in more detail.**

We agree with the reviewer that this information should be explained in more detail. We clarify that in the reference dataset, the intensive aerosol properties are first computed at the lidar's vertical resolution. Then, the average values for each aerosol layer are calculated and assigned uniformly to all height-levels within the corresponding layer. This results in a dataset where each layer contains both (i) the height-resolved values of intensive properties and (ii) the averaged properties repeated across all rows within the layer, as shown in Fig. 2. This approach is useful for ML models, as it allows them to learn from both the fine-scale variability of optical properties and the overall average characteristics of each aerosol type. The combination of detailed profiles and layer-averaged values helps the model associate typical aerosol types with their characteristic vertical distributions, while also promoting consistency across the layers. Also, we kindly refer to our previous response to the General Comment #4 for further explanations of the differences between height-resolved data and layer-averaged data, as well as the inclusion of clarifications in the revised version of the manuscript.

**b) Also, specify what of the two data representations is used as the reference dataset for comparison with the outputs of each ML method (page 4, lines 115-117).**

We agree with the reviewer that this should be clarified. For the first attempt of the annotations, we use the average values of the layers but after establishing the final annotation we apply the typing at high-resolution, which is then used as input for the ML models. Thus, we have included a clarification in line 119 of the revised version of the manuscript as follows:

"Based on the literature (Gross et al., 2013; 2014; Navas-Guzman et al., 2013; Illingworth et al., 2015; Soupiona et al., 2020), we assigned a single type of aerosol to each aerosol layer at average resolution, according to certain ranges of the calculated average intensive properties"

Additional clarifications are included in line 141 of the revised version of the manuscript to specify the information about which representation is used for comparison with the outputs of each ML method:

"Finally, the height-resolved labels from the reference dataset are used for comparison with the outputs of each ML method."

**c) The vertical resolution of the lidar profiles used in the reference database for the UGR station is indicated but not their time resolution (page 4, lines 92-94). This should be included.**

We agree with the reviewer that this information should be included. Thus, we have included it in line 92 of the revised version of the manuscript as follows:

"… with a vertical resolution of 7.5 m for half-hour profiles."

**2) Sect. 2.6: Please, provide more information on:**

**a) Table 1: Why did you applied those ranges of values for each hyperparameter?**

We thank the reviewer for this comment. The ranges of hyperparameter values have been shown in multiple previous studies that they cover a broad range of optimal hyperparameters values for each algorithm (e.g. Philippus et al., 2024). This is a common practice in the literature to reduce the excessive computational cost of the hyperparameter selection process.  We have added a brief explanation in the revised manuscript in section 2.6 (lines 203-204) to clarify this point:

"The selected ranges of hyperparameters were based on similar studies of the literature (e.g. Philippus et al., 2024; see Table 3 for evaluated values)."

Reference included in the revised manuscript:

"Philippus, D., Sytsma, A., Rust, A., and Hogue, T. S.: A machine learning model for estimating the temperature of small rivers using satellite-based spatial data, Remote Sensing of Environment, 311, 114271, https://doi.org/10.1016/j.rse.2024.114271, 2024."

**b) Page 8, lines 212-214: Which are the weights to each class applied?**

We agree with the reviewer that this should be clarified. The weights assigned to each class are proportional to their representation in the reference dataset, as stated in the original and now in the revised manuscript in lines 234-236: "To account for the imbalance in the distribution of aerosol classes, weighted metrics were used, which assign weights to each class, that are proportional to their representation in the reference dataset". However, we have included the formal notation to provide a better explanation. We have included the following information on lines 243-249 of the revised manuscript:

"The weights used in the calculation of the weighted metrics (such as weighted precision, recall, and F1-score) are based on the relative support of each class in the reference dataset. That is, each class is weighted proportionally to its number of instances in the dataset and is calculated as:

$$M_{weighted} = \sum_{i=1}^{C} \frac{n_i}{N} \cdot M_i$$

Where $M$ is the metric (precision, recall or F1-score), $C$ is the number of classes (aerosol types), $n_i$ is the number of samples in class $i$, $N$ is the total number of samples, and $M_i$ is the metric for each class $i$. These weighted metrics were used specifically to mitigate the influence of class imbalance in the evaluation."

**3) Sect. 3.1: Figures 3 and 4, and Table 2 (pages 10-11): They can be affected depending on the manual aerosol classification performed to differentiate the 'Unknown' type from the rest of aerosols, and between each aerosol class. Please, clarify.**

We thank the reviewer for this comment. The indicated figures and table by the reviewer, show a description of the reference dataset. The manual aerosol classification is a warrant that the aerosol typing has been revised by experts in the field and is of great quality, so that it is employed as "the ground-truth" for the supervised ML algorithms. In the case that other dataset would have been analyzed in this work, there would have been different representations in Figures 3 and 4, and Table 2, but the data analyzed in this paper has provided us with the results currently shown.

Regarding to the concern of the reviewer about the classification performed to differentiate the "Unknown" type or the rest of aerosol classes, we believe that the typing criteria has been clarified in General Comment #1, where it is shown that all the types that were not classified or did not had physical-meaning, were included as the "Unknown" class. Further information on the typing criteria is also included in Section 2.1 (please refer to our response to General Comment #1).

**4) Sect. 3.2:**

**a) Page 12: Table 3 could be merged with Table 1.**

We agree with the reviewer on this point. Thus, we have merged Table 1 into Table 3. The corresponding comments on Table 1 now refer to Table 3, with the updated table numbering in the revised manuscript. We have updated the table of Section 3.2 and the caption of the merged table is now as follows in lines 306-307 of the revised version of the manuscript:

"**Table 3**. Ranges of hyperparameters evaluated for each ML model and optimal hyperparameters selected for the different configurations."

| ML model | Hyperparameters evaluated | Configuration: with depolarization | Configuration: without depolarization |
|---|---|---|---|
| Decision Tree | max_depth: None, 10, 20, 30 min_samples_split: 2, 10, 20 | max_depth: None min_samples_split: 2 | max_depth: None min_samples_split: 2 |
| Random Forest | max_depth: None, 10, 20 n_estimators: 100, 200 | max_depth: 10 n_estimators: 100 | max_depth: 10 n_estimators: 200 |
| Gradient Boosting | learning_rate: 0.01, 0.1, 0.2 n_estimators: 100, 200 | learning_rate: 0.1 n_estimators: 200 | learning_rate: 0.2 n_estimators: 100 |
| XGBoost | learning_rate: 0.01, 0.1, 0.2 max_depth: 3, 6, 9 n_estimators: 100, 200 | learning_rate: 0.2 max_depth: 9 n_estimators: 200 | learning_rate: 0.2 max_depth: 6 n_estimators: 200 |
| LightGBM | learning_rate: 0.01, 0.1, 0.2 n_estimators: 100, 200 num_leaves: 31, 50, 100 | learning_rate: 0.2 n_estimators: 100 num_leaves: 31 | learning_rate: 0.2 n_estimators: 100 num_leaves: 31 |
| Neural Network | alpha: 0.0001, 0.001, 0.01 hidden_layer_sizes: (64, 64), (128,64), (128, 128) learning_rate_init: 0.001, 0.01 | alpha: 0.001 hidden_layer_sizes: (64, 64) learning_rate_init: 0.01 | alpha: 0.01 hidden_layer_sizes: (128, 128) learning_rate_init: 0.01 |

We have also removed the sentence "The configurations tested for all ML models are summarized in Table 1" and have included a reference to Table 3 in line 204 of Section 2.5 of the revised manuscript as:

"The selected ranges of hyperparameters were based on similar studies of the literature (e.g. Philippus et al., 2024; see Table 3 for evaluated values)."

We have also included additional information when describing Table 3 in lines 301-302 of Section 3.2 as follows:

"Table 3 summarizes the set of hyperparameters evaluated and provides the optimal hyperparameters for each ML model and for both configurations: (1) with depolarization and (2) without depolarization."

**b) Page 12, lines 281-282: It is not clear if the 416 layers examined in this work correspond to the reference dataset or the testing dataset. Please, clarify.**

We agree with the reviewer that this should be clarified. The 416 layers examined in this work correspond to the reference dataset. Thus, we have clarified this information in line 311 of the revised version of the manuscript by specifying that these 416 layers are part of the reference dataset:

"Six ML models were evaluated on the 416 layers from the reference dataset for the period 2012-2015, assessing all metrics on the test dataset for two configurations: (1) with depolarization and (2) without depolarization."

**c) Pages 12-13, lines 284-285: In the statement: 'In general, the ML models that incorporate depolarization data demonstrate significantly higher performance compared to those without depolarization', it should be added 'except for the neural network', as shown in Figure 5.**

We thank the reviewer for noticing this missing information and thus, we have included "except for the neural network" at the end of the sentence in line 315 of the revised manuscript.

**d) Page 13, lines 288-289: In the sentence: 'This might be explained due to a stronger influence of other features rather than depolarization on the aerosol classification problem with the NN setup'. Explain what do you mean by 'other features', please.**

We thank the reviewer for this comment. By "other features," we simply meant "the remaining features" in the dataset, excluding depolarization. These features might have a stronger influence on the aerosol classification problem when using the NN setup. We have modified the sentence accordingly in line 319 of the revised manuscript to improve clarity.

**e) Page 16, lines 362-365: In the statement, 'The Continental Polluted and Mixed types show comparable performances near ~70% in both configurations, indicating that these aerosol types are more difficult to be predicted in the classification process. They are also confused with Smoke in a ~30%, which makes sense due to nature of the Mixed aerosol type, which can be composed of several types of aerosols, being Smoke one of the major contributors.', why Smoke is one of the major contributors in the Mixed type? Maybe that is related to the way the Mixed class was identified? Please, clarify it.**

We agree with the reviewer that this should be clarified. We did not want to mean that Smoke must physically be the major contributor to Mixed class, but that the observed intensive properties for Smoke (according to literature and used for the initial manual annotation) presented large ranges (LR from 26 to 100) and linear depolarization ration (0.001 to 0.14). Those ranges overlap with the Mixed and Continental Polluted ranges, and this is also why the initial classification based on those ranges had to be corrected by experts and supported by backward

trajectories and aerosol concentration models. In this sense, it could be expected a certain degree of confusion by the ML model among those types. Indeed, the Mixed class was less confused (around 12%) with Smoke type in the scenario with depolarization data, probably due to the smaller overlap between these classes. We have clarified this point in the revised manuscript by adding a more detailed explanation in lines 402-411 of the revised manuscript:

"The Continental Polluted and Mixed types show similar performances (~70%) in both configurations, suggesting that these aerosol types are more challenging to predict in the classification process. This is partly due to the broad and overlapping ranges of lidar properties (e.g., LR and particle depolarization ratio) used for initial labelling, which can lead to confusion between classes such as Mixed, Continental Polluted, and Smoke. Although the Mixed type does not necessarily imply a physical dominance of Smoke, the optical properties associated with Smoke (e.g., lidar ratio from 26 to 100 sr, linear depolarization ratio from 0.001 to 0.14 in Table 1) can overlap with those of other classes, making separation difficult. Thus, a thorough expert review and the integration of additional information (e.g., backward trajectories and aerosol concentration models) were necessary to refine the final labels. Indeed, the Mixed class was less confused (around 12%) with Smoke type in the scenario with depolarization data, probably due to the smaller overlap between these classes."

**5) Sect. 3.3:**

**a) Page 17, line 397: Please, provide more detail for '… particularly when several intensive properties are missing, …'. Why?**

We thank the reviewer for pointing this out. By looking at Figure 8 of the original manuscript, we observe that when part of the data is missing, for example of the extinction profile in the bottom and top layers, and the LR also at those layers, the ML model tends to classify those layers as Unknown class. Thus, we have updated the phrase in line 460 of the revised version in order to better clarify this:

"…, particularly when several intensive properties lack of information, …"

**b) Pages 18-19, lines 413-418: Please, check this paragraph, there is an inconsistency. If there are not available depolarization information in the EARLINET database during the Saharan dust period, how the depolarization data could be included for the evaluation of the LightGBM performance in this configuration? Please, clarify.**

We thank the reviewer for pointing this potential inconsistency. Indeed, depolarization data is not available in the EARLINET database for the Saharan dust period. However, our research group has access to the depolarization data for this event, which was used to evaluate the performance of the LightGBM model in the configuration with depolarization. We have clarified this information in the revised manuscript in line 471 as follows:

"Depolarization information was unavailable in the EARLINET database during that period; however, depolarization data was provided by the research group for the same event. As a result, the NATALI algorithm …".

**6) Discussion Sect.:**

**a) Page 19, lines 431-434: Although it was mentioned in this section, the manual aerosol classification is not described in the work, and it can affect the reference dataset used.**

**See General Comment #1.**

We agree with the reviewer that the manual aerosol classification could have been explained in more detailed in the original work. Thus, thanks to the General Comment #1, we have improved

this description and as also included additional classifications in our response of the Specific Comment (Comment #3).

**b) Page 19, lines 435-439: See General Comment #4.**

We thank the reviewer for this comment and General Comment #4. We have already provided an answer in detail in the General Comment #4.

**Other minor comments:**

**1) Page 5, line 121: What do you mean with 'corresponding height of each layer'? Please,**

**specify.**

We thank the reviewer for pointing this out. The sentence was not fully clear. Previously it was: "For the cases where the aerosol type was not clear enough, we computed the backtrajectories with 120 HYSPLIT for 5 days in advanced and for the corresponding height of each layer.". Thus, we have clarified it in lines 125-128 of the revised version of the manuscript as follows:

"For the cases where the aerosol type was not clear enough, we computed the backtrajectories with the HYSPLIT model, taking as the starting point of the altitude for the backtrajectory analysis the one of each identified aerosol layer. Thus, for uncertain aerosol layer types, we ran the model for 5 days in advance to trace the air mass origin for the altitude corresponding to each layer."

**2) Page 5, line 128: What is the meaning of 'row' in this sentence?**

We thank the reviewer for this comment. The meaning of row in this sentence means altitude. Thus, we have replaced that word by "altitude" in line 140 of the revised version of the manuscript.

**3) Page 5, line 135: Why do you use the median value for the missing values? Explain.**

We thank the reviewer for this comment. In this study, the missing values of each variable at vertical resolution have been imputed with the median value of each profile. Hence, the median was chosen because it is less sensitive to extreme outliers compared to the mean, making it a more robust estimator for the central tendency of the data, especially when dealing with potential anomalies or skewed distributions in vertical profiles.

**4) Page 5, line 136: If each profile is denoted by $j$, does it have sense $j$=0? Also, define $\Delta z$.**

We thank the reviewer for pointing this out. The use of $j = 0$ is intentional, as we assigning variables to the profiles. In many programming languages, including Python, it is standard to index from 0. This allows for consistency with the way data structures, such as arrays or lists, are typically indexed in Python. However, to improve clarity, we have made a slight adjustment in the notation: we now use $n$-1, $m$-1, and $l$-1 instead of $n$, $m$, and $l$, where $n$, $m$, and $l$ refer to the number of datasets. In addition, we agree with the reviewer that $\Delta z$ should be defined. Thus, the following changes are included in the revised version of the manuscript:

In lines 149-150:

"Let us name each profile by $j = 0,1,2,\ldots,n-1$ and define the height discretization $z_i^j = z_0^j + i\Delta z$, $i = 0,1,2,\ldots,m-1$, where $n,m$ are positive integer values. $\Delta z$ represents the step in altitude, i.e. the vertical spatial resolution, and $z_i^j$ denotes that $z_0^j$ depends on the profile."

In line 184:

"Where $p_{i,k}^j$ represents the reference dataset, with all the variables $k = 0,1,\ldots,l-1$ from the dataset (Fig. 2), and $\hat{c}_i^j$ is the predicted class for each layer and height."

**5) Page 6, lines 170-172: If $k$ denotes the variables, $k = 0$ should be avoided, shouldn't it? Also, the term 'the predicted class for each layer and height' is correct? Either layer or height?**

Regarding the first question, it is already answer in the previous Comment 4 and the corresponding changes are made. Regarding the second question, we agree that the sentence might be confusing. The predicted classes are performed to each layer at height resolution. In order to avoid confusion, we have updated the sentence in lines 185 of the revised version of the manuscript:

"…, and $\hat{c}_i^j$ is the predicted class for each layer at height resolution."

**6) Page 7, line 198: Please define 'GroupKFold' (and include reference).**

We agree with the reviewer that this information should be included and referenced. Thus, we have included the information in lines 212-215 of the revised version of the manuscript:

"GroupKFold is a cross-validation strategy in which the data is split into training and testing sets while ensuring that the same group is not represented in both sets. The detailed explanation is included in the scikit-learn documentation on GroupKFold ([https://scikit-learn.org/stable/modules/generated/sklearn.model_selection.GroupKFold.html](https://scikit-learn.org/stable/modules/generated/sklearn.model_selection.GroupKFold.html))."

Reference included in the revised manuscript:

scikit-learn developers: GroupKFold, [https://scikit-learn.org/stable/modules/generated/sklearn.model_selection.GroupKFold.html](https://scikit-learn.org/stable/modules/generated/sklearn.model_selection.GroupKFold.html), last access: 15 May 2025.

**7) Page 7, line 198: Please define 'GroupShuffleSplit' (and include reference).**

Following the previous answer, we have also included the information about GroupShuffleSplit in lines 227-231 of the revised version of the manuscript and the corresponding reference:

"GroupShuffleSplit is a cross-validation strategy in which the data is split into training and testing sets while ensuring that the same group is not present in both sets, similar to GroupKFold. However, unlike GroupKFold, GroupShuffleSplit randomly shuffles the groups and allows for more flexible splits, where each group is assigned to either the training or testing set ([https://scikit-learn.org/stable/modules/generated/sklearn.model_selection.GroupShuffleSplit.html](https://scikit-learn.org/stable/modules/generated/sklearn.model_selection.GroupShuffleSplit.html))."

Reference included in the revised manuscript:

scikit-learn developers: GroupShuffleSplit, [https://scikit-learn.org/stable/modules/generated/sklearn.model_selection.GroupShuffleSplit.html](https://scikit-learn.org/stable/modules/generated/sklearn.model_selection.GroupShuffleSplit.html), last access: 15 May 2025.

**8) Page 10, Figure 3's caption: 'ratio' is missing at the end of 'particle linear depolarization'.**

Done.

**9) Page 3, Figure 5's caption: Should it be 'in the testing dataset' instead of 'in the reference dataset'?**

We thank the reviewer for noticing this discrepancy. We have made the modification and updated Figure's 5 caption to read "in the testing dataset".

**10) Page 14, Figure 6: The title in the colour bar may be 'Feature importance' instead of 'Feature value'?**

We thank the reviewer for her/his suggestion. The title "Feature value" is used in the colour bar of Figure 6 because it represents the actual values of the features (variables) in the model, with higher values in red and lower values in blue, with respect to their SHAP value (impact on the model output). In contrast, "Feature importance" refers to the relative contribution of each feature to the model's overall predictions, which is a different concept. Since the plot displays the actual feature values for each feature (rather than their importance) in the colour bar, we believe the current title is more accurate in this context. In addition, the features are ordered by their importance, as explained in the previous version of the manuscript and detailed in lines 348-350 of the revised version of the manuscript.

**11) Page 17, line 371: Check: ' … which was used for neither training nor testing the model, …'.**

We thank the reviewer for this comment.  We have updated the sentence to: "… which were used for neither training nor testing the model.", as recommended, in line 416 of the revised version of the manuscript.

**12) Page 18, Figure 8: Check the x-axis: the title of the first panel, and also the scale in the second panel (it should be multiplied by $10^6$).**

We thank the reviewer for pointing this out. We have implemented all the requested changes in the figure of the revised version of the manuscript, which currently corresponds to Figure 9.

**13) Pages 21-28: Check the reference list as wrong and missing details are found.**

We thank the reviewer for this comment. We have carefully checked the reference list and we have made the following changes:

- We have substituted the surname "Gross" by "Groß" in the revised manuscript, as the latter is used for referencing. In addition, the correct year of this reference was 2014 (as stated in the reference list) but in the manuscript was written 2015. Therefore, we have modified the year with 2014 in lines 39 and 118 of the revised manuscript.
- Regarding the reference Belegante et al., 2014, the year was mistaken in the reference list and in the text of the manuscript. We have now corrected an updated it in the revised manuscript in line 104 and in the reference list.
- The reference of Alabadla et al. (2022), Breiman and Spector (1992) and Matus et al. (2019) were not in the Copernicus format in the reference list. Thus, we have updated them and they are as follows in the revised manuscript:

Alabadla, M., Sidi, F., Ishak, I., Ibrahim, H., Affendey, L. S., Che Ani, Z., Jabar, M. A., Bukar, U. A., Devaraj, N. K., Muda, A. S., Tharek, A., Omar, N., and Jaya, M. I. M.: Systematic Review of Using Machine Learning in Imputing Missing Values, IEEE Access, 10, 44483–44502, https://doi.org/10.1109/access.2022.3160841, 2022.

Breiman, L. and Spector, P.: Submodel Selection and Evaluation in Regression. The X-Random Case, International Statistical Review / Revue Internationale de Statistique, 60, 291, https://doi.org/10.2307/1403680, 1992.

Matus, A. V., L'Ecuyer, T. S., and Henderson, D. S.: New Estimates of Aerosol Direct Radiative Effects and Forcing From A-Train Satellite Observations, Geophysical Research Letters, 46, 8338–8346, https://doi.org/10.1029/2019gl083656, 2019.

- The following reference was missing at the reference list and has been included in the revised version of the manuscript:

Soupiona, O., Samaras, S., Ortiz-Amezcua, P., Böckmann, C., Papayannis, A., Moreira, G. A., Benavent-Oltra, J. A., Guerrero-Rascado, J. L., Bedoya-Velásquez, A. E., Olmo, F. J., Román,

R., Kokkalis, P., Mylonaki, M., Alados-Arboledas, L., Papanikolaou, C. A., and Foskinis, R.: Retrieval of optical and microphysical properties of transported Saharan dust over Athens and Granada based on multi-wavelength Raman lidar measurements: Study of the mixing processes, Atmospheric Environment, 214, 116824, https://doi.org/10.1016/j.atmosenv.2019.116824, 2019.

We would like to thank again the reviewer for all the valuable comments that indeed have improved the manuscript.

**Response to Referee #2 – del Águila et al.:**

We would like to thank the reviewer for the positive comments that definitely have helped to improve the manuscript. In the following, a point-by-point response to the reviewer's comments is included below. Reviewer's comments are written in bold while the answers to each comment are written in regular format. Changes in the manuscript are noted between quotation marks ("") and underlined, and are referred to the corresponding line in the revised version of the manuscript.

**This paper presents a very innovative and relevant study, showing the possibility of using ML techniques to predict the type of aerosols. The work is very well written and structured. However, the following points need to be better detailed.**

We thank again the reviewer for the valuable comments that have improved the manuscript.

**1) Line 135: Why was the median used to fill in the gaps? Did you try to use other techniques? Perhaps the use of machine learning techniques could generate a more robust filling.**

We thank the reviewer for the insightful suggestions. The first question was already answered to Reviewer #1 (Minor Comment #3). Thus, in this study, the missing values of each variable at vertical resolution have been imputed with the median value of each profile because the median is less sensitive to extreme outliers compared to the mean, making it a more robust estimator for the central tendency of the data, especially when dealing with potential anomalies or skewed distributions in vertical profiles. We also tested the mean for imputation but found no significant improvement in the results.

We also agree that exploring alternative imputation techniques, such as machine learning (ML)-based approaches (e.g., K-Nearest Neighbors, regression models, or deep learning methods), could provide more robust imputations, especially in datasets with complex relationships or patterns. However, it is important to consider that these techniques could introduce another layer of uncertainty, as they would depend on the assumptions made during model training and the quality of the data used for learning. For instance, ML imputation methods might overfit to noise or introduce biases if the model does not generalize well to unseen data. Due to these potential sources of uncertainty, we chose to use the median imputation, as it offers a simpler, more stable approach with less risk of introducing additional complexity into the analysis. Nonetheless, we agree that exploring these more advanced techniques could be a valuable approach for future research.

**2) Table 1: Why were these groups of hyperparameters exclusively selected? Was any analysis of the importance of hyperparameters performed? This can severely affect the final performance of the models, especially for neural networks.**

We understand the reviewer's questions. The first question was also raised by Reviewer #1 (Specific Comment #2a). In general, the groups of hyperparameters selected have been shown in multiple previous studies to cover a broad range of optimal hyperparameter values for each algorithm (e.g. Philippus et al., 2024). This approach is common practice in the literature to reduce the excessive computational cost associated with an exhaustive hyperparameter search.

While we did not perform an explicit analysis of the individual importance of each hyperparameter, the chosen ranges are based on well-established values that have been validated through various studies, ensuring that the selected configurations are representative of well-performing models. Hyperparameter optimization via GridSearchCV was used for these models to search within these ranges. We recognize that the selection of hyperparameters is

crucial and might affect the model performance, especially for neural networks (NN), and we believe that the chosen hyperparameter ranges are suitable for our study's objectives while balancing computational efficiency. Finally, we have included one sentence to the revised version of the manuscript to provide a clearer explanation in section 2.6, lines (203-205):

"The selected ranges of hyperparameters were based on similar studies from the literature (e.g. Philippus et al., 2024; see Table 3 for evaluated values)."

Reference:

Philippus, D., Sytsma, A., Rust, A., and Hogue, T. S.: A machine learning model for estimating the temperature of small rivers using satellite-based spatial data, Remote Sensing of Environment, 311, 114271, https://doi.org/10.1016/j.rse.2024.114271, 2024.

**3) Figure 4: I recommend increasing the font size of the axes.**

We agree with the reviewer on this point. In the revised manuscript, we have updated the figure by increasing the font to size 18 for both axes labels and size 16 for the tick labels.

**4) Figure 4: How did you deal with the problem of the imbalance of the dataset? Because "continental polluted" tends to have worse performance due to the smaller number of data.**

We thank the reviewer for her/his comment. We agree that the imbalance in the dataset, particularly for the Continental Polluted, can affect the model performance. As we clarified in our response to Reviewer #1 (Specific Comment #2b), we addressed class imbalance in the evaluation phase by using weighted metrics, where each class contributes proportionally to its representation in the reference dataset (see Section 2.6, lines 234-236). A clarification regarding the dealing with the imbalanced class has been included in 2.6 in lines 248-249 of the revised manuscript as follows:

"These weighted metrics were used specifically to mitigate the influence of class imbalance in the evaluation."

**5) Line 285: I expected better results from the NN model with depolarization data, since you have more information about the particle analyzed. Isn't this difference associated with the data input format in the model? Was any preprocessing performed to normalize them?**

We thank the reviewer for her/his comment. We acknowledge that the inclusion of depolarization data should theoretically provide more information about the particles and might enhance the NN model performance, but this is highly dependent on how the model processes and uses this additional data. The performance discrepancy between NN models with and without depolarization is not necessarily related to the input format, but rather depends on several factors, including the complexity of the data and the way the NN model handles the additional information.

Regarding the specific line commented by the reviewer, mentioning the study of Nicolae et al. (2018), it is important to note that in their study, depolarization adds more discriminative information, which allows for a greater number of aerosol categories to be identified, but it does not necessarily imply a direct increase in classification accuracy when depolarization is taken into account. Additionally, we believe the reviewer may be referring to the comparison with NATALI, but it is important to highlight that our approach differs significantly in terms of the input data. We are using height-resolved data, which has been preprocessed differently compared to the data used in NATALI, that trained the NN with simulated data. These differences in data preparation and resolution may explain the variation in performance between the models,

making a direct comparison challenging. We had already included this information in the original version of the manuscript, which is currently in lines 477-481 of the revised manuscript. Finally, we would like to clarify that we did not normalize the data before inputting it into the NN, which indeed could have influenced the model's performance. This decision was made to maintain consistency with the other ML models tested, ensuring the use of the same dataset across all models. However, we appreciate the reviewer's suggestion and will consider normalization as an option for future studies.

**6) Figure 5: Considering the use by other users, I think it is important to comment on the computational cost of each model.**

We agree with the reviewer on this point. Thus, we have included a sentence regarding the computational costs of each model in lines 339-344 of Section 3.2.2 of the revised manuscript:

"To evaluate the computational costs of the models, we recorded the total computational time on a standard workstation equipped with a 12th Gen Intel(R) Core(TM) i7-12700H CPU and 32 GB of RAM. Simpler models such as Decision Trees and Random Forests completed training in under 20 seconds. More complex models like XGBoost and LightGBM required between 20 seconds and 1 minute. The Neural Network model required approximately 2 minutes. The Gradient Boosting model had the highest computational cost, with a training time of around 8 minutes. Despite these differences, once the models are trained, all models provide near-instantaneous predictions."

**7) Section 3.2.3: Was an analysis of multicollinearity between the features performed? This can affect the importance of each one in the model, as well as the performance of the final model.**

We thank the reviewer for this comment. We agree that multicollinearity between features can influence both the model's performance and the interpretation of feature importance.

In our study, we did perform a multicollinearity analysis using a feature correlation matrix, as shown in the correlation matrix included in Figure R1. This analysis showed that while some features are indeed correlated, as expected for physically related variables such as backscatter coefficients, extinction, and lidar ratios; many of the key input variables, including the most important ones identified by SHAP (e.g., averaged lidar ratio at 532 nm, depolarization ratio, and color index), do not exhibit strong mutual correlations with each other. Therefore, although moderate multicollinearity is present among some features, the SHAP results still reflect meaningful and distinct contributions from physically interpretable variables. This suggests that LightGBM, which is relatively robust to multicollinearity due to its tree-based structure and use of techniques like Exclusive Feature Bundling (EFB), is handling the redundancy effectively.

Nevertheless, we acknowledge that further analysis on collinearity or dimensionality reduction techniques could help refine the feature set and improve interpretability. We appreciate the reviewer's suggestion and will consider this in future studies.

[Figure]

*Figure R1. Heatmap of the correlation matrix for all the features of the reference dataset of the imputed lidar data.*

**8) Line 323: Because of this statement, I expected that depolarization would present better results in the MLP Classifier.**

We thank again for the reviewer's comment. The statement that the reviewer is referring to is the following: "The results highlight the relevance of the physical information of the intensive properties (e.g. CI, CR, LR, AE) in the general aerosol classification problem." We understand the expectation that the inclusion of depolarization, which is a key intensive property widely used in traditional aerosol classification, should also result in better model performance, particularly for neural networks such as MLPs. As we previously noted in Comment #5, the inclusion of depolarization does indeed provide additional physical information about aerosol particles. However, the actual improvement in model performance depends not only on the presence of informative variables but also on how effectively the model architecture can learn and generalize from this information, particularly when the dataset is complex and high-dimensional, as in our case. In our study, we observed that the MLP classifier did not outperform other models when depolarization was included. This can be attributed to several interrelated factors such as:

a) Input data characteristics: Our model is trained on height-resolved lidar profiles, which significantly increases the input dimensionality compared to the simpler, layer-averaged data commonly used in other studies (e.g., Nicolae et al., 2018). This higher dimensionality requires careful regularization and large training datasets to fully leverage the additional features without overfitting.

b) Data preprocessing: As previously mentioned, no normalization was applied to the input features, in order to maintain consistency across all ML models evaluated. While tree-based models are less sensitive to feature scaling, NN are known to benefit from normalized inputs.

This lack of normalization may have limited the MLP's ability to learn from the depolarization values effectively.

c) Model architecture: The MLP architecture used in this study was kept relatively simple to allow fair comparisons across models and avoid excessive computational costs. However, more complex architectures (e.g., deeper neural networks or convolutional models) might be better suited to exploit the additional information provided by depolarization and vertical structure.

Therefore, while the SHAP analysis highlights the physical importance of depolarization as a feature (as stated in Line 323 of the original manuscript), this does not automatically translate into better performance for all model types, as shown in the MLPs, unless the input data and model architecture are specifically optimized for that purpose.

Finally, we will clarify this point in the manuscript and will consider including more advanced preprocessing and architectural adaptations in future work. We have included in lines 357-360 of the revised manuscript the following explanation:

"It should be noted that although depolarization appears as a highly relevant feature, this does not automatically lead to improved performance for all model types, as occurred with the MLP classifier (Figure 5), because the improvement depends on how each algorithm processes and learns from the available information."

**9) Line 363: I recommend reviewing the imbalanced dataset issue because if this is not corrected, the cases that are less present in the training tend to perform worse.**

We thank the reviewer for this comment. We fully agree that class imbalance can negatively impact the classification performance of underrepresented classes, such as Continental Polluted. As mentioned in our previous response to Comment #4, we addressed this issue by applying class weights during model evaluation, which allowed us to emphasize minority classes when assessing the model's performance.

In relation to the specific sentence on Line 363 of the original manuscript: "The Continental Polluted and Mixed types show comparable performances near ~70% in both configurations, indicating that these aerosol types are more difficult to be predicted in the classification process", we agree that the statement could benefit from further clarification. As detailed in our response to Reviewer #1 in Specific Comment #4e, the Continental Polluted and Mixed classes present overlapping lidar properties (e.g., lidar ratio and linear depolarization ratio), which increases the difficulty in distinguishing them from other aerosol types such as Smoke. This, combined with their lower representation in the dataset, contributes to a lower classification performance of these classes. However, the model achieves an accuracy of around 70% for both classes, which is considered a reasonable result given the inherent complexity of aerosol classification in lidar observations.

We have revised the manuscript accordingly to clarify the factors influencing the model's performance for these classes, as detailed in the response to Reviewer #1 in Specific Comment #4e, where we included the following lines 402–411 in the revised manuscript:

"The Continental Polluted and Mixed types show similar performances (~70%) in both configurations, suggesting that these aerosol types are more challenging to predict in the classification process. This is partly due to the broad and overlapping ranges of lidar properties (e.g., LR and particle depolarization ratio) used for initial labelling, which can lead to confusion

between classes such as Mixed, Continental Polluted, and Smoke. Although the Mixed type does not necessarily imply a physical dominance of Smoke, the optical properties associated with Smoke (e.g., lidar ratio from 26 to 100 sr, linear depolarization ratio from 0.001 to 0.14 in Table 1) can overlap with those of other classes, making separation difficult. Thus, a thorough expert review and the integration of additional information (e.g., backward trajectories and aerosol concentration models) were necessary to refine the final labels. Indeed, the Mixed class was less confused (around 12%) with Smoke type in the scenario with depolarization data, probably due to the smaller overlap between these classes."